# Next-generation phenotyping integrated in a national framework for patients with ultrarare disorders improves genetic diagnostics and yields new molecular findings

Individuals with ultrarare disorders pose a structural challenge for healthcare systems since expert clinical knowledge is required to establish diagnoses. In TRANSLATE NAMSE, a 3-year prospective study, we evaluated a novel diagnostic concept based on multidisciplinary expertise in Germany. Here we present the systematic investigation of the phenotypic and molecular genetic data of 1,577 patients who had undergone exome sequencing and were partially analyzed with next-generation phenotyping approaches. Molecular genetic diagnoses were established in 32% of the patients totaling 370 distinct molecular genetic causes, most with prevalence below 1:50,000. During the diagnostic process, 34 novel and 23 candidate genotype–phenotype associations were identified, mainly in individuals with neurodevelopmental disorders. Sequencing data of the subcohort that consented to computer-assisted analysis of their facial images with GestaltMatcher could be prioritized more efficiently compared with approaches based solely on clinical features and molecular scores. Our study demonstrates the synergy of using next-generation sequencing and phenotyping for diagnosing ultrarare diseases in routine healthcare and discovering novel etiologies by multidisciplinary teams.

A recent analysis of the Orphanet database showed that around 3–6% of the global population have a rare disease (that is, a disease with a prevalence of <1 in 2,000) and that 72% of such cases may have a genetic cause[1]. Rare diseases thus represent a substantial global health burden. However, only a minority of patients suspected to have a rare disease receive both a definite clinical diagnosis and a confirmatory molecular test result[2,3]. This concerns in particular the subset of patients with ultrarare disorders that are defined in the European Union as affecting no more than one person in 50,000 and that follow a long tail distribution with respect to their frequency (Regulation (EU) No. 536/2014). It is estimated that roughly 80% of

the more than 5,000 rare genetic diseases have a prevalence below one in a million[1].

The International Rare Disease Research Consortium therefore stated that, by 2027, all patients who come to medical attention with a suspected rare or ultrarare disease should be diagnosed within 1 year if the respective disorder has been described in the medical literature[4]. Since many rare diseases are Mendelian in nature, comprehensive genetic testing is a key element to achieve that goal.

In Germany, around 90% of the population has statutory health insurance, and the current reimbursement scheme allows physicians to request chromosome analyses, molecular karyotyping and

✉e-mail: pkrawitz@uni-bonn.de

**Fig. 1 | Workflow in the TRANSLATE NAMSE project and phenotypes in which exome sequencing was performed. a**, Patients with a suspected rare disease were referred to a MDT and deeply phenotyped using HPO terminology. If a genetic etiology was considered likely, exome sequencing was performed. The MDT then evaluated the molecular findings and could order additional analyses for variants of uncertain significance or variants in potentially novel disease candidate genes (created with BioRender.com). **b**, Exome sequencing was performed predominantly in children. The main indications for exome sequencing in children were neurodevelopmental disorders. In adults, the main indications were neurological/neuromuscular disorders. In both children and adults, the least common disease categories were 'cardiovascular',

'endocrine, metabolic, mitochondrial, nutritional' (emmn) and 'hematopoiesis/immune system' (his). **c**, Phenotypic similarities between patients, as encoded according to their HPO terms, were visualized with UMAP. As reference, all OMIM diseases were included using their HPO annotations (gray background dots). For each patient, color coding indicates allocation to disease groups, in accordance with the leading clinical feature. An overlap is evident for patients in the neurodevelopmental and neuromuscular groups (aquamarine and blue clusters), which indicates high phenotypic similarity. This precludes the unequivocal assignment of these patients to a diagnostic group. The triangles indicate patients who contributed to the identification of a novel, high-evidence gene–phenotype association.

sequencing of single genes or gene panels. For example, high-resolution genome-wide array-based segmental aneusomy profiling detects a pathogenic aberration in around 19% of patients with developmental delay[5]. Besides contiguous gene syndromes, most of the remaining rare disorders are monogenic and are caused by single nucleotide variants or small insertions or deletions (indels). However, single gene analyses or small gene panels are only likely to detect a pathogenic aberration if the phenotype is highly predictive of the molecular cause, for example, hemoglobinopathies[6].

For phenotypes with high genetic heterogeneity, such as neurodevelopmental disorders, genetic investigation is more challenging. For intellectual disability, for example, studies so far have identified disease associations for more than a thousand genes[7]. For these disorders, research has shown that exome sequencing can be more cost-effective than sequencing potentially multiple gene panels[8]. However, this is also accompanied by more genetic variants that have to be assessed. Therefore, a clear indication for exome sequencing and efficient data analysis strategies are crucial. Between 2018 and 2020, a novel diagnostic concept within the German healthcare system was evaluated in the prospective study TRANSLATE NAMSE[9].

This involved standardized structures and procedures and multidisciplinary teams (MDTs) at ten university hospital-based centers for rare diseases (CRDs). The MDTs conducted a three-step diagnostic process: (1) primary review of patient records; (2) selection of diagnostic procedures, including a possible recommendation for exome sequencing; and (3) evaluation of all findings, including genetic variants. A key goal was to investigate whether exome sequencing would facilitate the diagnosis of ultrarare disorders or even the delineation of novel monogenic disorders. In this work, we report the molecular findings of this study.

Furthermore, we investigated how phenotypic features can be used to estimate the probability that a molecular diagnosis can be established with exome sequencing (YieldPred). In a companion study, we also assessed the extent to which the results from computer-assisted pattern recognition in facial dysmorphism contribute to variant interpretation (prioritization of exome data by image analysis, PEDIA). The present analyses demonstrated that exome sequencing facilitated the diagnosis of ultrarare genetic diseases and novel gene–disease associations and that artificial intelligence (AI)-driven technologies improved the diagnostic yield for ultrarare genetic disorders.

## Results

### Phenotypic characteristics of the study cohort

Between 2018 and 2020, a total of 5,652 individuals (2,033 adults and 3,619 children) with a suspected rare disorder were enrolled in TRANSLATE NAMSE by CRDs at ten German university hospitals (Fig. 1a)[9]. The present analyses were performed using the data from a total of 1,577 of these 5,652 patients (268 adults, 1,309 children). In these individuals, the MDT at the respective CRD considered a genetic cause as plausible and exome sequencing as the most suitable test (exome sequencing cohort, Supplementary Table 1). Each of these 1,577 individuals was assigned to one of six major disease categories by the respective CRD physician (Fig. 1b). The majority of children were assigned to the disease category 'neurodevelopmental disorders' (n = 702, 54%), and the largest proportion of adults were assigned to the disease category 'neurological or neuromuscular disorders' (n = 117, 44%). Smaller proportions of adult and pediatric cases were assigned to the groups 'organ malformation', 'endocrine/metabolic disorders', 'immune/hematologic disorders' and 'cardiovascular disorders'. Patient phenotypes were also annotated with terms of the Human Phenotype Ontology (HPO) by the respective CRD physicians. On average, five HPO terms were specified per individual (Supplementary Fig. 1a). The phenotypes within the present cohort were visualized by projecting the patient-specific HPO terms into a two-dimensional space. While most patients from the same disease

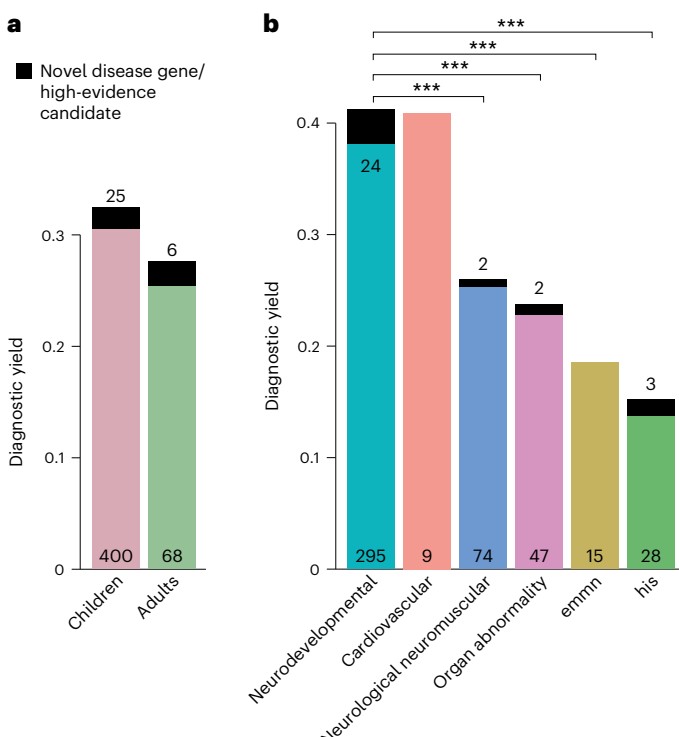

**Fig. 2 | Diagnostic yield of exome sequencing depends on age and disease group. a,b**, The diagnostic yield differed according to age group (adult/child) (**a**) and disease category (**b**). For all disease categories, with the exception of cardiovascular, the diagnostic yield was increased by novel DGGs and high-evidence candidate genes (dark-colored tip of the bar). The absolute number of solved cases in which a variant was found in an established disease gene is given at the bottom of each bar, and the number of solved cases attributable to a novel DGG or high-evidence candidate gene is given at the top of each bar. The entire TRANSLATE NAMSE exome sequencing cohort was considered for **a** and **b** (n = 1,577). Diagnostic yield between disease categories were compared using two-sided Fisher's exact test. P values were adjusted by Bonferroni correction. ***P < 0.001; exact corrected P values: neurodevelopmental (ndd) versus neurologic neuromuscular P = 5.4 × 10⁻⁵, ndd versus organ abnormality P = 5.2 × 10⁻⁵, ndd versus emmn P = 5.9 × 10⁻⁴, ndd versus his P = 1.1 × 10⁻¹¹. emmn, endocrine, metabolic, mitochondrial, nutritional; his, hematopoiesis/immune system.

group were in close proximity, the clusters showed a partial overlap (Fig. 1c). For example, many patients categorized within 'neurological or neuromuscular disorders' also showed HPO terms typically associated with 'neurodevelopmental disorders' and vice versa (Supplementary Fig. 1b). This suggests that grouping patients into single disease groups may be overly simplistic.

### Diagnostic yield of exome sequencing

A molecular diagnosis was established in a total of 499 of the 1,577 patients (32%), that is, in these cases, exome sequencing identified variants that fully or partially explained the phenotype. The diagnostic yield was slightly higher in children (32%) than in adults (28%, P = 0.13, Fisher's exact test; Fig. 2a) and twofold higher in patients assigned to the category 'neurodevelopmental disorder' than for all other disease categories (42% versus 22%, P < 0.001, Fisher's exact test with Bonferroni correction; for single comparisons between disorder groups, see Fig. 2b). Furthermore, exome sequencing found variants of uncertain significance. Specifically, these variants were enriched for missense variants (80% versus 45%, P < 0.001; Supplementary Fig. 2), due to lower support for pathogenicity according to the guidelines of the American College of Medical Genetics (ACMG) and the Association for Molecular Pathologists for interpretation of sequence variants.

## De novo variants and parental mosaicism

A total of 228 diagnoses (45% of 510 diagnoses including dual diagnoses) were attributable to de novo variants, making them the most common cause of disease in families with an autozygosity below 0.02 and the second most common cause in families with consanguinity (Fig. 3). In three families with variants that were initially classified as de novo, evidence for probable or certain parental mosaicism was found (Supplementary Note). In one of these families, the same likely pathogenic variant in *PUF60* was identified as the cause of developmental delay in two affected brothers. Since the variant was not detectable in the exome data of either parent, gonadal mosaicism could not be confirmed and was instead presumed on the basis of the family history. The detection in the exome sequencing analysis of three probable parental mosaics among 228 patients corresponds to a frequency of 1.3%, which is within the estimated interval of clinically relevant parental mosaicism[10–12].

## Recessive disease burden

The second-largest proportion of solved cases involved an autosomal recessive (AR) mode of inheritance (125 solved cases, 14.5% of all diagnoses; Fig. 3a). In total, 94 of the causative variants in the 125 recessive diagnoses in the present cohort would also have been classified as pathogenic if identified in healthy individuals[13]. The diagnostic yield was considerably higher in patients with presumed consanguinity (low autozygosity 31%, $n = 1,014$ versus high autozygosity 41%, $n = 144$, $P = 0.01$, Fisher's exact test), and the composition of the modes of inheritance also differed significantly between the high- and low-autozygosity groups (Fig. 3b). The relative contribution of homozygous variants was significantly higher in the high-autozygosity group (73% of $n = 62$ diagnoses) than in the low-autozygosity group (2% of $n = 313$ diagnoses) (odds ratio (OR) 111.5, $P < 0.001$, Fisher's exact test). In contrast, the contribution to disease of de novo variants was 13% ($n = 62$ diagnoses) in the high-autozygosity group compared with 54% ($n = 313$ diagnoses) in the low-autozygosity group (OR 0.2, $P < 0.001$, Fisher's exact test). Since the de novo mutation count is dependent on parental age but not on autozygosity, the disease prevalence that is attributable to de novo variants should be comparable between both groups and can be used for normalization (Fig. 3c). For an inbreeding coefficient of >2%, this suggests a recessive disease burden that is sevenfold higher than for those with lower inbreeding coefficients, which is consistent with previous reports[14–16]. However, it also has to be acknowledged that population expansion results in a drop in the prevalence of recessive disorders in random mating populations and that the lower recessive disease burden might be only a transient effect[17].

**Fig. 3 | Mode of inheritance and disease burden are dependent on autozygosity. a**, Pie chart showing the distribution of modes of inheritance (MOI) for all diagnoses ($n = 510$). Most disease-causing variants occurred de novo and on an autosome. At least 75% of all autosomal recessive diagnoses could have been identified by expanded carrier screening (slice). **b**, Box plots of autozygosity for each MOI ($n = 375$). Individuals are indicated by gray dots. Autozygosity was substantially increased in individuals with autosomal recessive disorders due to homozygous variants. In the box plots, the center lines indicate the median values, and the bottom and top edges of the boxes are the first (25%) and the third (75%) quartiles. The whiskers extend to the minimal and maximal data points with a maximum distance of 1.5 interquartile ranges from the edges of the box. **c**, Bar graphs illustrating MOI in individuals with low (<2%, $n = 313$) and high (>2%, $n = 62$) autozygosity. On the right, the autosomal dominant de novo rate has been used for normalization. Individuals with high autozygosity had a higher relative burden of recessive diseases, mainly due to the presence of homozygous pathogenic variants. The box plots present the median as the center line, the upper and lower quartiles as box limits, and 1.5× the interquartile range as the whisker length (in the style of Tukey). AD, autosomal dominant inheritance, variant inherited or of unknown origin; AD (de novo), autosomal dominant inheritance with de novo variant; AR (comp het), autosomal recessive inheritance with compound heterozygous variants; AR (hom), autosomal recessive inheritance with homozygous variant; mt, mitochondrial inheritance; XL, X-linked inheritance.

## Dual molecular diagnoses and secondary findings

For 11 individuals, who represented approximately 2% of all solved cases, molecular diagnoses for two distinct or overlapping disease phenotypes were established (Supplementary Table 2). This group showed a tendency for high autozygosity (43%, $n = 7$ versus 16%, $n = 361$, $P = 0.09$, Fisher's exact test) and recessive disorders (41%, $n = 22$ diagnoses versus 24%, $n = 488$ diagnoses, $P = 0.08$, Fisher's exact test). The detected percentage of dual diagnoses (2%, 11 of 499 solved cases) is consistent with both the enrichment of high autozygosity and recessive disorders in this group, and earlier reports[18,19].

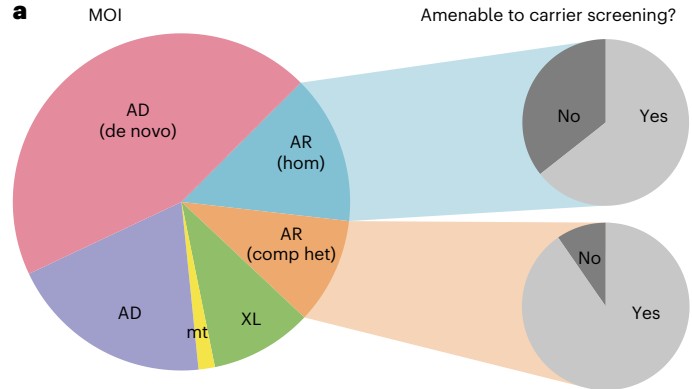

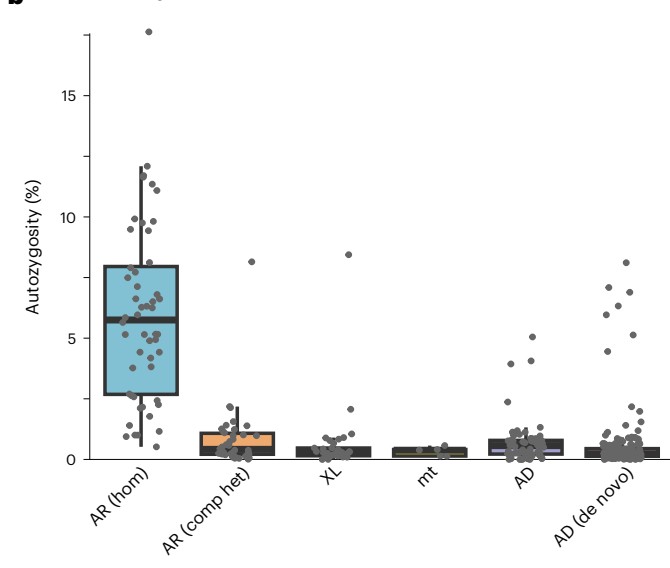

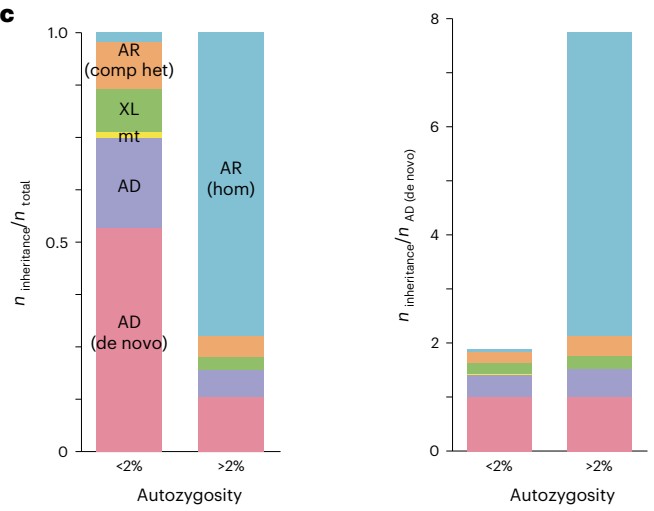

In 17 individuals who had consented to being informed about secondary findings, we identified medically actionable variants that were unrelated to the present phenotype. The list of 59 actionable genes was based on the ACMG recommendations; however, secondary findings in 7 additional genes were reported following discussions within the respective MDTs (Supplementary Note).

## Enrichment of ultrarare diagnoses

For the 499 individuals in whom exome sequencing led to a molecular diagnosis, a total of 549 disease-causing variants were identified in 362 different disease-associated genes as well as structural variants affecting 14 genomic regions (Supplementary Table 1). This plethora of diagnoses suggests that each specific genetic disorder had a very low prevalence. To clarify this, the results were compared with the total number of (likely) pathogenic ClinVar submissions for the respective genes (Fig. 4a). The first quartile of ClinVar variants corresponds to the more frequently identified rare diseases and contains 40,078 variants assigned to 47 genes. In the group of 499 individuals with a molecular diagnosis in the present cohort, only 33 patients and 14 different disease-associated genes fell into this first quartile. In contrast, the majority of the present 499 patients (corresponding to 192 different disorders) were assigned to the fourth quartile, which contains disease genes with the least ClinVar submissions (Fig. 4b). Notably, almost half of the diagnoses assigned in the present cohort were only established in the past decade (Fig. 4c). A comparison with a cohort of comparable size[20] revealed a significantly different distribution with respect to the years in which the phenotype was first associated with the respective disease-causing gene (Kolmogorov–Smirnoff test, $P < 0.001$; Supplementary Figs. 3 and 4).

## Novel DGGs and candidates

In cases for which no molecular diagnosis could be established due to variants in the known clinical exome, all potentially deleterious variants in the remaining exome were assessed for plausible novel disease etiologies (see detailed scoring for 57 candidate genes in 65 cases in Methods, Supplementary Note and Supplementary Table 3). Moderate evidence was generated for 23 of 57 candidate genes, and high evidence was generated for the remaining 34. A total of 17 candidate genes with high evidence are currently undergoing further investigation, mostly within the framework of international projects. A total of 17 genes (12 with autosomal dominant inheritance, 5 with autosomal recessive inheritance) have acquired diagnostic-grade gene (DGG) status during the first three years through international cooperation[21–33]. After the end of the study, two more candidate genes transitioned to the group of DGGs due to additional phenotypic, functional and statistical evidence became available[32,34].

In comparison with pathogenic variants in previously known disease-associated genes, the present candidate gene set showed a higher proportion of missense variants. This is probably attributable to the fact that the classification of missense variants is more challenging (Supplementary Table 3).

## Functional assays

For 18 cases that were classified as uncertain or unsolved after initial exome sequencing, multi-omic assays were performed, that is, an analysis of the methylome ($n = 4$), proteome ($n = 3$) or transcriptome ($n = 14$). Epigenetic signatures, as derived from methylome analyses, clarified the status of de novo missense variants as likely benign in one case and as pathogenic in three. This is exemplified by a case with a missense variant in *KMT2D* (Supplementary Note)[35,36]. Variants in *MDH2* were reclassified to pathogenic, on the basis of a proteome analysis of patient-derived fibroblasts (Supplementary Note), while results were inconclusive in two unsolved cases. In 13 unsolved cases, RNA sequencing was performed but could not identify transcriptome alterations that lead to the identification of causative variants. Thus, in 5/18 cases,

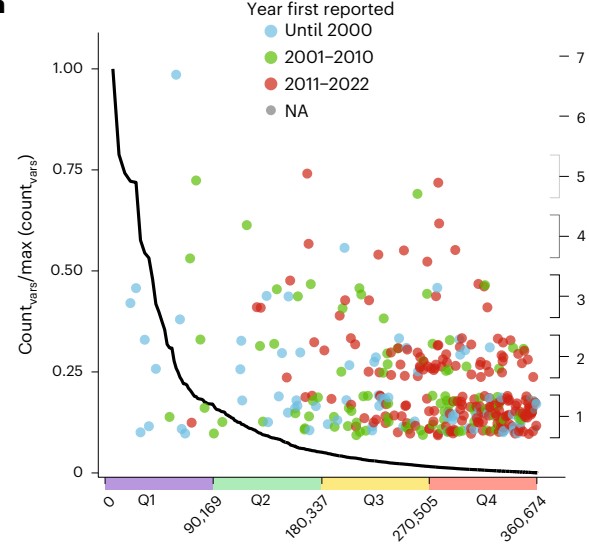

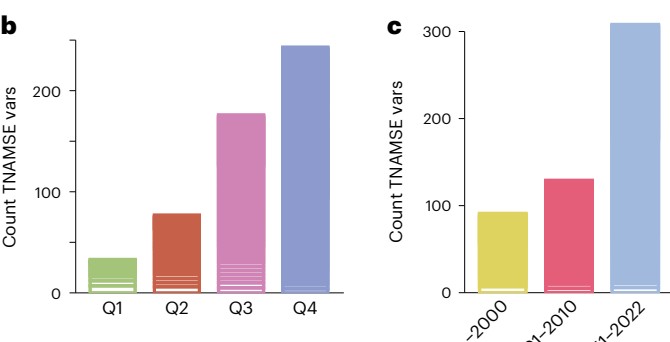

**Fig. 4 | Most variants identified in TRANSLATE NAMSE exome sequencing cohort cause ultrarare disorders that were first associated with a gene in the last decade. a**, Comparison of the number of (likely) pathogenic variants per gene in TRANSLATE NAMSE relative to the frequency of submission of (likely) pathogenic variants to ClinVar. Genes are ordered from left to right according to a decreasing frequency of ClinVar submissions. The black line corresponds to the complementary cumulative distribution (1 – CDF; cumulative distribution function) of ClinVar submissions. Diagnostic variants in TRANSLATE NAMSE (counts displayed on the right axis) were plotted as dots above their respective gene and in the color corresponding to the year in which the gene was first described as being associated with the respective disease. **b**, Variant counts in TRANSLATE NAMSE in genes with high (first quartile, Q1) to low (Q4) counts of submissions per gene in ClinVar. The genes in Q1–Q4 each cover approximately 1/4 of the submissions of likely or confirmed pathogenic variants to ClinVar, as shown on the *x* axis in **a**. Variants in the same gene are grouped in horizontal blocks. **c**, Bar graph showing the number of variants relative to the time interval in which the gene was first described as being associated with the respective disease. Note that 59 genes listed in the recommendations for reporting of secondary findings (version 2) of the ACMG were excluded from the analyses to counteract potential biases in ClinVar due to submissions of secondary findings[67]. TNAMSE, TRANSLATE NAMSE; vars, variants.

complementary assays facilitated variant reclassification and highlighted the importance of variant validation strategies in diagnostics for suspected rare genetic diseases (Supplementary Note)[37–39].

## Predicting the diagnostic yield using machine learning

Analyses were then conducted to investigate whether the phenotype predicted the diagnostic yield of exome sequencing. For this purpose, a least absolute shrinkage and selection operator (LASSO) analysis for binary outcomes was performed. To reduce the phenotypic dimension

and to increase interpretability, HPO terms were first aggregated into 49 nonoverlapping phenotypic groups. These phenotypic groups were used as predictors in the LASSO analysis. The resulting model was able to discriminate between solved and unsolved cases (Supplementary Fig. 5a; area under the curve (AUC) 0.67, 95% confidence interval (CI) 0.61–0.74, on a held-out test set of the exome sequencing cohort, $n = 321$) and yielded the HPO groups 'dysfunction of higher cognitive abilities', 'hematological abnormalities' and 'ataxia' as very influential predictors in terms of the establishment of a molecular diagnosis via exome sequencing (Fig. 5a). To improve the predictions for a wider variety of phenotypic features, we trained on samples of additional cohorts and made the model available as a web service (https://translate-namse.de). YieldPred can now be used to estimate the diagnostic yield of exome sequencing on the basis of the phenotypic features of a given patient and might therefore help in expectation management (Methods and Supplementary Figs. 3, 5 and 6).

### Variant prioritization using facial image analysis (PEDIA)

A total of 224 of the 1,577 patients had also provided written informed consent for the evaluation of their facial images with the AI tool Gestalt-Matcher[40] and the use of the results (gestalt scores) in exome variant interpretation (PEDIA)[41]. In 94 of these PEDIA subcohort cases, a molecular diagnosis was established. For 81 of these 94 cases, the gestalt scores improved prioritization results, that is, the correct diagnosis was ranked higher. In general, the PEDIA approach (that is, a combined scoring approach involving genotype-, phenotype- and facial gestalt-based prioritization tools) can contribute to prioritization efficiency, provided that (1) the clinical features of the underlying disorder include facial dysmorphism and (2) molecularly solved cases are already part of the GestaltMatcher Database[40] (https://db.gestaltmatcher.org/). In the present PEDIA subcohort, for 81 cases, representing 68 different disorders, one or more previously solved cases were phenotypically so similar that the gestalt score for the associated disease gene resulted in a higher ranking for the pathogenic variant than prioritization approaches that do not make use of image analysis.

Four different variant prioritization approaches involving genotype-based and/or phenotype-based scores were analyzed and their respective accuracy rates compared. For the PEDIA approach, the correct disease-associated gene was listed among the top ten suggestions in 82% of the cases. The PEDIA approach outperformed prioritization by either a molecular score (combined annotation-dependent depletion, CADD[42]) or GestaltMatcher only, as well as the combined molecular and feature score (CADD + case annotation and disorder annotation (CADA)) (Fig. 5b). As the latter can be considered routine in exome sequencing analysis, additional gestalt scores help to improve variant interpretation in diagnostics.

Based on these results and the extension of the TRANSLATE NAMSE study beyond the initial 3 years, the PEDIA workflow was implemented at further sites. The exome sequencing data of another 149 patients were then analyzed. In this additional cohort, a molecular diagnosis was established in 69 patients, and a top-10 accuracy of 83% was achieved using the PEDIA score (Supplementary Fig. 7).

The PEDIA approach is highly modular, and the GestaltMatcher score for image analysis can also be combined with other prioritization tools such as Exomiser[43], Xrare[44], LIRICAL[45] or Amelie[46], which use different molecular scores or HPO-based scores. All tested combinations showed improvements in the top-$k$ accuracies and are discussed in Supplementary Note and Supplementary Fig. 8.

In some cases, the gestalt scores were particularly suggestive and facilitated the identification of otherwise challenging pathogenic variants. For instance, in a patient with a very high gestalt score for Koolen de Vries syndrome, a 4.7-kb de novo deletion affecting *KANSL1* was detected[47]. Other case reports of particular interest are described in Supplementary Note and Supplementary Fig. 9.

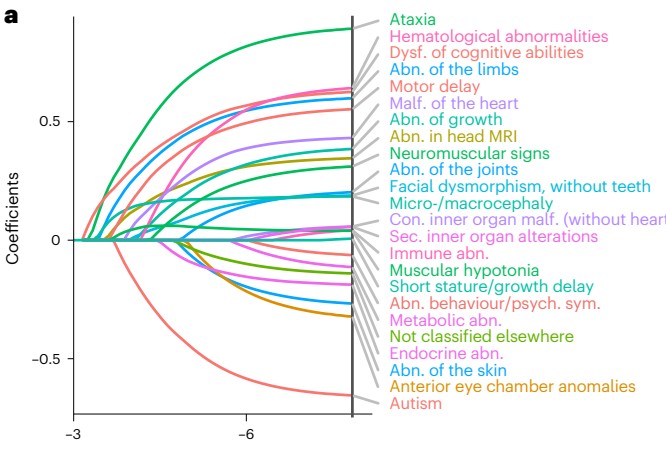

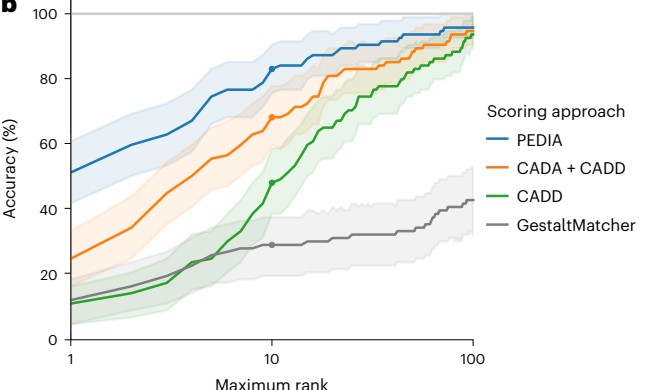

**Fig. 5 | Machine learning identifies features relevant to the diagnostic yield and can support variant prioritization. a**, The coefficient paths of regression analysis using the LASSO are shown. Only features that are included in the final model and are present in at least 5% of the cases that were used for training are depicted. The more to the left [lower ln(λ)] a coefficient path starts to deviate from the *x* axis, the more informative the corresponding feature is in terms of predicting the diagnostic yield. Features with positive coefficients increase the diagnostic yield. In contrast, features with negative coefficients render a monogenic cause less likely. For example, dysfunction of higher cognitive abilities and ataxia are associated with a higher diagnostic yield (clinical features are colored according to their higher-order HPO groups; for details, see Supplementary Note). An algorithm to predict the diagnostic yield (YieldPred) was developed on the basis of these data and can be found online (https://translate-namse.de). **b**, The performances of variant prioritization approaches were compared. All disease-associated genes were ranked using the respective variant prioritization method. Subsequently, the proportion of cases detected with the correct disease-associated gene (sensitivity) was shown as a function of the number of disease-associated genes considered, beginning at the top score. The following four approaches for variant prioritization were tested in solved cases from the PEDIA cohort ($n = 94$): (1) only a molecular pathogenicity score (CADD[68]) with top-10 accuracy of 48%; (2) feature-based score (CADA[69]) in addition to CADD with top-10 accuracy of 68%; and (3 and 4) a gestalt score from facial image analysis (GestaltMatcher[40]) alone or in addition to both CADD and CADA referred to as PEDIA score[41] with top-10 accuracy of 82%. Note that the bold lines indicate the observed top-$k$ accuracy and bootstrapped 95% CIs are indicated by the lighter shading around the lines. MRI, magnetic resonance imaging; abn., abnormality; con., congenital; dysf., dysfunction; psych., psychiatric; sym., symptoms; sec, secondary.

### Exemplary diagnoses with targeted therapy

Implications of diagnoses on clinical management were not assessed in a structured way. However, for five patients in the TRANSLATE NAMSE cohort with a molecular diagnosis (1%), individualized treatments or therapies directed against the mechanism of the disease could be initiated[48]. A patient with metachromatic leukodystrophy due to pathogenic

variants in arylsulfatase alpha was treated with autologous CD34+ cells that were transduced ex vivo using a lentiviral vector encoding aryl-sulfatase alpha[49]. The gene therapeutic approach with atidarsagene autotemcel has been authorized by European Medicines Agency (EMA) in the European Union since 17 December 2020. A patient with pyruvate dehydrogenase E1-α deficiency due to a de novo variant in *PDHA1* and another patient with GLUT1-deficiency due to pathogenic variants in *SLC2A1* were treated with a ketogenic diet. In a patient with cerebral creatine deficiency syndrome 1, due to a missense substitution in *SLC6A8*, supplementation with creatine was started. In a patient with congenital disorder of glycosylation of type IIc, due to a homozygous missense variant in *SLC35C1*, the fucosylation deficiency was treated by oral fucose supplementation[50].

## Discussion

Reducing the time to diagnosis from several years to less than 1 year is highly relevant in terms of both prognosis and the targeted use of healthcare resources, since the number of approved therapies for rare diseases in which early treatment is associated with better outcomes is now increasing[51]. Establishing a molecular diagnosis quickly will require the implementation of frameworks within healthcare systems that are dedicated to patients with rare diseases. The novel diagnostic approach evaluated in TRANSLATE NAMSE was the practical realization of such a concept. The present investigation suggests that a combination of a structured clinical assessment by an MDT, an advanced sequencing test, such as exome sequencing, and a comprehensive discussion of the results reduces diagnostic delay and may improve therapy. These findings are consistent with reports from other healthcare systems and other disorders that benefit from interdisciplinary structures[20,52–56]. On the basis of the present data, in 2021, exome sequencing was included in the list of standard medical services offered to patients with suspected rare diseases who were referred to German CRDs. For all the patients that are still awaiting a molecular diagnosis, new multi-omics approaches are promising but also costly. Therefore, in a complex healthcare system, these tests compete with other analyses, and their efficiency and efficacy in establishing a diagnosis should be evaluated in the future. However, it will be crucial within the German healthcare system that the inclusion of MDTs in the diagnostic process does not delay or even hinder genetic testing for patients with rare diseases. With exome sequencing being incorporated into an increasing number of guidelines, we also anticipate that the focus of the MDT will shift from test selection toward variant interpretation and identifying therapeutic options. By these means, MDTs operating in CRDs would fulfill a similar purpose for patients with rare disorders as molecular tumor boards in centers for personalized medicine already do for cancer patients[57].

Two notable findings of the present analyses were that, in comparison with ClinVar and a previously reported rare disease cohort of similar size[20], the TRANSLATE NAMSE cohort was significantly enriched for ultrarare disorders (Fig. 4a and Supplementary Fig. 4) and that a large number of recently described gene–disease associations were found[1,8,20,58]. In our opinion, this accumulation of ultrarare diagnoses and the relative absence of more common conditions is explained by the study protocol, which required consideration of different test options, including gene panels. Furthermore, the fact that a large number of the established diagnoses have only become possible in recent years as a result of increasing medical genetic knowledge (Fig. 4c) highlights the importance of reanalysis of exome data[59,60]. Indeed, the present analyses identified a large number of individuals who carried variants that indicated a novel disease–gene association (12% of solved cases), which highlights the fact that the analysis of exome sequencing data should not be limited to known disease genes. Establishing novel gene–disease associations and conducting functional analyses for the reclassification of variants of uncertain significance are time-consuming and highly complex endeavors[61]. Hence, from the present logistical perspective, such analyses are easier to perform in a

research context than within the routine diagnostic context of clinical practice. However, these findings are of crucial importance for affected individuals and their families. Thus, from a teleological perspective, in some rare disease cases, boundaries separating diagnostics and research are somewhat blurred. Therefore, in the tertiary, academic setting, collaboration between experts from diagnostics and research is highly relevant for patients with suspected ultrarare diseases and a lack of definitive diagnostic findings.

In several patients from the present cohort, molecular diagnoses also resulted in a change of clinical management to a causal or even curative approach to therapy as described above. These cases emphasize the fact that molecular genetic diagnoses are essential in terms of the development of personalized treatments or therapies that are directed against the underlying disease mechanism. The systematic, consortium-based collection of molecular and clinical data represents the first necessary milestone toward achieving this goal. Particularly in the case of ultrarare disorders, the collection of these data requires additional international collaborative efforts.

Besides the ability to select the appropriate genetic test for diagnosing a disease, a core competence of a clinical geneticist is to estimate disease risk in the offspring of healthy individuals.[17] In addition to the relatedness of the partners, the burden of heterozygous pathogenic variants in recessive genes, which can vary considerably depending on demographics[62–65], could play an increasingly important role in family planning. In a total of 94 of the 125 cases with recessive molecular diagnoses, the causal variants would also have been classified as (likely) pathogenic if they had been identified in healthy individuals[13]. This also means that, if the parents of pediatric patients with a recessive disorder in the present cohort had undergone exome sequencing to determine their carrier status, three out of four of these couples could have received appropriate genetic counseling concerning disease risk in future offspring, which supports the argument for extended screening[66].

Another aim of the present study was to determine whether complementary AI and machine learning approaches would facilitate diagnostic effectiveness and efficiency in the exome sequencing cohort. The PEDIA analyses showed that AI-powered next-generation phenotyping increased the efficiency of exome sequencing data analysis. However, not every case in the present cohort was solved via exome sequencing. Therefore, the machine learning model YieldPred was developed to identify features that had a major impact on the diagnostic yield in our and other study cohorts. Prospectively, this approach can also be used for two purposes. First, it can be used to estimate the probability that exome sequencing will result in a molecular diagnosis in each patient with a suspected rare disease and can by these means help to manage expectations. Second, as YieldPred in its current form provides an estimation of the diagnostic yield of exome sequencing and not of an underlying monogenic condition of a certain individual, it can be used to stratify individuals for more comprehensive genetic testing, that is, a low YieldPred score despite a high likelihood of a monogenic disease indicates that transcriptomics, proteomics or genome sequencing could be promising.

It would be desirable for all individuals with a suspected monogenic disorder for whom no definitive diagnosis can yet be established to have the option of participating in large-scale genomic diagnostic and research initiatives. We present TRANSLATE NAMSE as the German framework that organizes diagnostics for patients with ultrarare diseases with a backbone of case conferences in MDTs in academic CRDs. TRANSLATE NAMSE represents the first national-level project for undiagnosed patients in Germany, and the future expansion of the network on both the national and international level is planned.

In summary, the results of the present study demonstrate that our novel, structured diagnostic concept facilitates the identification of ultrarare disorders on a national level, provides undiagnosed patients with the opportunity to participate in international research,

and represents a platform for data sharing that facilitates the development of machine learning and AI tools to improve the diagnostic yield.

## Online content

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

**Axel Schmidt** [1,54], **Magdalena Danyel** [2,3,54], **Kathrin Grundmann** [4,54], **Theresa Brunet** [5,54], **Hannah Klinkhammer** [6,7], **Tzung-Chien Hsieh** [6], **Hartmut Engels** [1], **Sophia Peters** [1], **Alexej Knaus** [6], **Shahida Moosa** [8], **Luisa Averdunk** [9], **Felix Boschann** [2,3], **Henrike Lisa Sczakiel** [2,3], **Sarina Schwartzmann** [2], **Martin Atta Mensah** [2,3], **Jean Tori Pantel** [2,10], **Manuel Holtgrewe** [11], **Annemarie Bösch** [12], **Claudia Weiß** [12], **Natalie Weinhold** [12], **Aude-Annick Suter** [12], **Corinna Stoltenburg** [12], **Julia Neugebauer** [12], **Tillmann Kallinich** [12], **Angela M. Kaindl** [13,14,15], **Susanne Holzhauer** [12], **Christoph Bührer** [12], **Philip Bufler** [12], **Uwe Kornak** [2], **Claus-Eric Ott** [2], **Markus Schülke** [2], **Hoa Huu Phuc Nguyen** [16], **Sabine Hoffjan** [16], **Corinna Grasemann** [17], **Tobias Rothoeft** [17], **Folke Brinkmann** [17], **Nora Matar** [17], **Sugirthan Sivalingam** [1], **Claudia Perne** [1], **Elisabeth Mangold** [1], **Martina Kreiss** [1], **Kirsten Cremer** [1], **Regina C. Betz** [1], **Martin Mücke** [18], **Lorenz Grigull** [18], **Thomas Klockgether** [19], **Isabel Spier** [1],

André Heimbach[1], Tim Bender[18], Fabian Brand[6], Christiane Stieber[18], Alexandra Marzena Morawiec[18], Pantelis Karakostas[20], Valentin S. Schäfer[20], Sarah Bernsen[18], Patrick Weydt[19], Sergio Castro-Gomez[19], Ahmad Aziz[19], Marcus Grobe-Einsler[19], Okka Kimmich[19], Xenia Kobeleva[19], Demet Önder[19], Hellen Lesmann[1], Sheetal Kumar[1], Pawel Tacik[19], Meghna Ahuja Bhasin[6], Pietro Incardona[6], Min Ae Lee-Kirsch[21,22], Reinhard Berner[21,22], Catharina Schuetz[21,22], Julia Körholz[21,22], Tanita Kretschmer[21,22], Nataliya Di Donato[21,23], Evelin Schröck[21,23], André Heinen[21,22], Ulrike Reuner[21,24], Amalia-Mihaela Hanßke[21], Frank J. Kaiser[25], Eva Manka[26], Martin Munteanu[25], Alma Kuechler[25], Kiewert Cordula[26], Raphael Hirtz[26], Elena Schlapakow[27], Christian Schlein[28], Jasmin Lisfeld[28], Christian Kubisch[28,29], Theresia Herget[28], Maja Hempel[28,29,30], Christina Weiler-Normann[29,31], Kurt Ullrich[29], Christoph Schramm[29,31], Cornelia Rudolph[29], Franziska Rillig[29], Maximilian Groffmann[29], Ania Muntau[32], Alexandra Tibelius[30], Eva M. C. Schwaibold[30], Christian P. Schaaf[30], Michal Zawada[30], Lilian Kaufmann[30], Katrin Hinderhofer[30], Pamela M. Okun[33], Urania Kotzaeridou[33], Georg F. Hoffmann[33], Daniela Choukair[33], Markus Bettendorf[33], Malte Spielmann[34], Annekatrin Ripke[35], Martje Pauly[36,37], Alexander Münchau[35,38], Katja Lohmann[39], Irina Hüning[34], Britta Hanker[40], Tobias Bäumer[35,38], Rebecca Herzog[35,36], Yorck Hellenbroich[41], Dominik S. Westphal[5], Tim Strom[5], Reka Kovacs[5], Korbinian M. Riedhammer[5,42], Katharina Mayerhanser[5], Elisabeth Graf[5], Melanie Brugger[5], Julia Hoefele[5], Konrad Oexle[43], Nazanin Mirza-Schreiber[43], Riccardo Berutti[43], Ulrich Schatz[5], Martin Krenn[5,44], Christine Makowski[45], Heike Weigand[46], Sebastian Schröder[46], Meino Rohlfs[46], Katharina Vill[46], Fabian Hauck[46], Ingo Borggraefe[46], Wolfgang Müller-Felber[46], Ingo Kurth[10], Miriam Elbracht[10], Cordula Knopp[10], Matthias Begemann[10], Florian Kraft[10], Johannes R. Lemke[47,48], Julia Hentschel[47], Konrad Platzer[47], Vincent Strehlow[47], Rami Abou Jamra[47], Martin Kehrer[4], German Demidov[4], Stefanie Beck-Wödl[4], Holm Graessner[49], Marc Sturm[4], Lena Zeltner[49], Ludger J. Schöls[50], Janine Magg[49], Andrea Bevot[51], Christiane Kehrer[51], Nadja Kaiser[51], Ernest Turro[52], Denise Horn[2], Annette Grüters-Kieslich[53], Christoph Klein[46], Stefan Mundlos[2], Markus Nöthen[1], Olaf Riess[4], Thomas Meitinger[5], Heiko Krude[53], Peter M. Krawitz[6,55] ✉, Tobias Haack[4,55], Nadja Ehmke[2,3,55] & Matias Wagner[5,43,46,55]

[1]Institute of Human Genetics, University of Bonn, Medical Faculty and University Hospital Bonn, Bonn, Germany. [2]Institute for Medical Genetics and Human Genetics, Charité – Universitätsmedizin Berlin, Berlin, Germany. [3]BIH Charité Clinician Scientist Program, Berlin Institute of Health at Charité – Universitätsmedizin Berlin, Berlin, Germany. [4]Institute for Medical Genetics and Applied Genomics, University of Tübingen, Tübingen, Germany. [5]Institute of Human Genetics, Klinikum rechts der Isar, School of Medicine, Technical University of Munich, München, Germany. [6]Institute for Genomic Statistics and Bioinformatics, University of Bonn, Medical Faculty and University Hospital Bonn, Bonn, Germany. [7]Institut für Medizinische Biometrie, Informatik und Epidemiologie, University of Bonn, Medical Faculty and University Hospital Bonn, Bonn, Germany. [8]Institute for Medical Genetics, Stellenbosch University, Cape Town, South Africa. [9]Department of Pediatrics, University Hospital Düsseldorf, Düsseldorf, Germany. [10]Institute for Human Genetics and Genomic Medicine, Medical Faculty, Uniklinik RWTH Aachen University, Aachen, Germany. [11]Core Uni Bioinformatics, Berlin Institute of Health at Charité – Universitätsmedizin Berlin, Berlin, Germany. [12]Department of Pediatrics, Charité – Universitätsmedizin Berlin, Berlin, Germany. [13]Department of Pediatric Neurology, Charité – Universitätsmedizin Berlin, Berlin, Germany. [14]Center for Chronically Sick Children, Charité – Universitätsmedizin Berlin, Berlin, Germany. [15]Institute of Cell and Neurobiology, Charité – Universitätsmedizin Berlin, Berlin, Germany. [16]Department of Human Genetics, Ruhr University Bochum, Bochum, Germany. [17]Department of Pediatrics Bochum and CeSER, Ruhr University Bochum, Bochum, Germany. [18]Center for Rare Diseases, University of Bonn, Medical Faculty and University Hospital Bonn, Bonn, Germany. [19]Department of Neurology, University of Bonn, Medical Faculty and University Hospital Bonn, Bonn, Germany. [20]Clinic for Internal Medicine III, University of Bonn, Medical Faculty and University Hospital Bonn, Bonn, Germany. [21]University Center for Rare Diseases, University Hospital Carl Gustav Carus, Dresden, Germany. [22]Department of Pediatrics, University Hospital Carl Gustav Carus, Dresden, Germany. [23]Institute for Clinical Genetics, University Hospital Carl Gustav Carus, Dresden, Germany. [24]Department of Neurology, University Hospital Carl Gustav Carus, Dresden, Germany. [25]Institute of Human Genetics, University Hospital Essen, Essen, Germany. [26]Department of Pediatrics II, University Hospital Essen, Essen, Germany. [27]Department of Neurology, University Hospital Halle, Halle, Germany. [28]Institute of Human Genetics, University Hospital Hamburg-Eppendorf, Hamburg, Germany. [29]Martin Zeitz Center for Rare Diseases, University Hospital Hamburg-Eppendorf, Hamburg, Germany. [30]Institute of Human Genetics, Heidelberg University, Heidelberg, Germany. [31]I. Department of Medicine, University Hospital Hamburg-Eppendorf, Hamburg, Germany. [32]Department of Pediatrics, University Hospital Hamburg-Eppendorf, Hamburg, Germany. [33]Center for Child and Adolescent Medicine, University Hospital Heidelberg, Heidelberg, Germany. [34]Institute of Human Genetics, University Hospital Schleswig-Holstein, Lübeck, Germany. [35]Center for Rare Diseases, University Hospital Schleswig-Holstein, Lübeck, Germany. [36]Department of Neurology, University Hospital Schleswig-Holstein, Lübeck, Germany. [37]Institute for Neurogenetics, University Hospital Schleswig-Holstein, Lübeck, Germany. [38]Institute of Systems Motor Science, University of Lübeck, Lübeck, Germany. [39]Institute of Neurogenetics, University of Lübeck, Lübeck, Germany. [40]Institute of Human Genetics, University of Lübeck, Lübeck, Germany. [41]Department of Human Genetics, University Hospital Schleswig-Holstein, Lübeck, Germany. [42]Department of Nephrology, Klinikum rechts der Isar, School of Medicine, Technical University of Munich, München, Germany. [43]Institute of Neurogenomics, Helmholtz Zentrum München, München, Germany. [44]Department of Neurology, Medical University of Vienna, Wien, Austria. [45]Department of Paediatrics, Adolescent Medicine and Neonatology, München, Germany. [46]Dr. von Hauner Children's Hospital, University Hospital Munich, München, Germany. [47]Institute of Human Genetics, University of Leipzig Medical Center, Leipzig, Germany. [48]Center for Rare Diseases, University of Leipzig Medical Center, Leipzig, Germany. [49]Center for Rare Diseases, University of Tübingen, Tübingen, Germany. [50]Department of Neurology, University of Tübingen, Tübingen, Germany. [51]Department of Pediatric Neurology and Developmental Medicine, University of Tübingen, Tübingen, Germany. [52]Department of Genetics and Genomic Sciences, Icahn School of Medicine at Mount Sinai, New York, NY, USA. [53]Berlin Centre for Rare Diseases, Charité – Universitätsmedizin Berlin, Berlin, Germany. [54]These authors contributed equally: Axel Schmidt, Magdalena Danyel, Kathrin Grundmann, Theresa Brunet. [55]These authors jointly supervised this work: Peter M. Krawitz, Tobias Haack, Nadja Ehmke, Matias Wagner. ✉e-mail: pkrawitz@uni-bonn.de

## Methods

### Enrollment, research ethics and consent

A detailed description of the TRANSLATE NAMSE project is provided elsewhere[9,70]. In brief, participants for TRANSLATE NAMSE were recruited between January 2018 and December 2020 from a total of ten German CRDs (Berlin, Bochum, Bonn, Dresden, Duisburg/Essen, Hamburg, Heidelberg, Kiel/Lübeck, München and Tübingen). Overall coordination of the recruitment process was performed by the Institute of Public Health Berlin. This study is governed by the approval of the following institutional review boards: Charité – Universitätsmedizin Berlin, Germany (EA2/140/17); UKB Universitätsklinikum Bonn, Germany (Lfd.Nr.386/17); Universitätsklinikum Essen, University Duisburg-Essen, Germany (17-7774-BO); Universitätsklinikum Heidelberg, Germany (S-499/2017); Universitätsklinikum Tübingen, Germany (643/2017BO1); Universität zu Lübeck, Germany (17-272); Ludwig-Maximilians-Universität München, Germany (17-640); Ärztekammer Hamburg, Germany (MC-316/17); Technische Universität Dresden, Germany (AK 464122017). All patients or their legal guardians provided written informed consent before inclusion. The inclusion criteria for TRANSLATE NAMSE were the lack of a definitive diagnosis and the clinical suspicion of a rare disease. The medical records and family history of each individual were evaluated by a MDT, which comprised at least board-certified physicians of two specialities with domain-specific expertise. For each individual, the respective MDT then made recommendations concerning diagnostics and further clinical management. To make the recommendation of exome sequencing, a board-certified human geneticist was additionally required within the MDT. For example, strong criteria for the indication of exome sequencing were congenital malformations, a syndromic phenotype, a positive family history suggestive of a monogenic disease and lack of absence of an alternative test with a comparable suspected diagnostic yield. A total of 1,577 patients (268 adult and 1,309 pediatric) from the TRANSLATE NAMSE cohort were referred for exome sequencing on the recommendation of the MDT at the respective CRD (exome sequencing cohort). The phenotypic and molecular genetic data of these 1,577 patients were evaluated in the present analyses.

### Clinical and laboratory phenotype data

Clinical and laboratory phenotype data were transferred to the sequencing laboratory in the form of hard-copy case report forms or as online data capture applications (Face2Gene Clinic). Online data capture allowed the free entry of HPO terms. Data from hard-copy report forms and free-text entries were transformed into HPO terms. The phenotypes reported in the present study are those that were reported to the sequencing laboratories. On the basis of the leading presenting clinical feature, each case was assigned to one of six major disease groups (Supplementary Fig. 1b). This allowed a more definitive statement on diagnostic yield in relation to the clinical features of the patient. In the subsequent analyses, all assigned HPO terms ($n = 1,649$) were compiled and divided into higher-order groups ($n = 12$) and subcategories ($n = 49$) by expert clinicians. Therefore, patients were additionally assigned to at least one higher-order group as well as at least one subgroup. To assign a patient to an HPO-defined group, the patient had to have at least one of the HPO terms belonging to the respective group. The following higher-order groups were defined: 1, neurodevelopmental; 2, neuromuscular; 3, seizures; 4, growth disorders; 5, facial dysmorphism; 6, abnormality of connective tissue; 7, congenital malformations; 8, endocrine and metabolic abnormalities; 9, immune and hematological abnormalities; 10, sensory organ alterations; 11, abnormal findings on brain magnetic resonance imaging; 12, others. Within the respective higher-order groups, HPO terms were further assigned to subcategories ($n = 49$) (https://github.com/Ax-Sch/TNAMSE_geno_pheno/blob/main/resources/hpo_categorization_19_12_2022.tsv).

### DNA sequencing

Details on DNA sequencing for each sequencing laboratory are given in Supplementary Table 4. Trio sequencing was conducted for 58% of the cases. When additional informative relatives were available, these were also included in the analysis as permitted by German law (healthy minors were not analyzed). EDTA-treated whole-blood samples or saliva kits were delivered to one of the five participating sequencing centers (Berlin, Bonn, LMU Munich, Munich or Tuebingen) for further processing. After DNA extraction, fragment size and purity were assessed. If the DNA fulfilled all quality criteria, the sample was submitted for sequencing. Exome sequencing was performed on exon targets that were isolated using capture and either Agilent SureSelect Human All Exon kits v6 or v7 (Agilent Technologies), or the Human Core Exome Kit (Twist Bioscience). One microgram of DNA was sheared into 350–400-bp fragments, which were then repaired, ligated to adaptors and purified for subsequent polymerase chain reaction amplification. Amplified products were then captured by biotinylated RNA library baits in solution, in accordance with the manufacturer's instructions. Bound DNA was isolated with streptavidin-coated beads and reamplified. The final isolated products were sequenced using the Illumina NextSeq 500, NextSeq 550, HiSeq 2500 or NovaSeq 6000 sequencing system and 2 × 100-bp paired-end reads (Illumina). All five sequencing centers ensured a coverage of over 20× in over 95% of the RefSeq target region.

### Exome sequencing data-processing pipeline

Details on exome sequencing data processing for each sequencing laboratory are given in Supplementary Table 4. At each of the five sequencing centers, exome sequencing processing pipelines were established according to best practice guidelines. The DNA sequence was mapped to the published human genome build GRCh37 reference sequence using Burrows–Wheeler Aligner (BWA). The most up-to-date version at the time of sequencing was used, progressing from BWA v0.7.11 through to BWA-Mem v0.7.17[71,72]. Single nucleotide variants and small indels were detected with HaplotypeCaller (v3.7, v3.8 or v4.1; three laboratories, 40.0% of cases), Freebayes (v1.2.0, one laboratory, 16.6% of cases) or HaplotypeCaller as well as SAMtools v.0.1.7 (one laboratory, 43.4% of cases)[73,74]. Mitochondrial DNA variants were assessed using data from exome sequencing in three laboratories (80% of cases)[75]. Copy number variations were detected using ExomeDepth or ClinCNN on short-read data (two laboratories, 60.0% of cases), before exome sequencing by array CGH (two laboratories, 30.0% of cases) or not evaluated (one laboratory, 10.1%)[76,77]. Additionally, analysis for structural variants was only conducted by one laboratory (16.6% of cases). Analysis for uniparental disomy was performed in two sequencing laboratories (60.0% of cases) using the UpdHunter function of ngs-bits v2019_09 (https://github.com/imgag/ngs-bits) or custom scripts. Finally, analyses for mosaic variants were conducted by four laboratories (90% of cases).

Variants were annotated using VEP (four laboratories, 80.2% of cases)[78] or Jannovar (one laboratory, 19.8% of cases)[79] and analyzed in VarFish[80], megaSAP (https://github.com/imgag/megSAP) or EVAdb (https://github.com/mri-ihg/EVAdb) or in tabular format depending on the center. Virtual gene panels were used in four out of five sequencing sites (56.7% of cases). In the sequencing site where no virtual panels were used, a similar approach (HPO-based and Online Mendelian Inheritance in Man (OMIM) full-text search) was used. Additionally, filter parameters specific for assumed modes of inheritances were applied (all laboratories; mainly cutoffs of allele frequencies or counts in the population database gnomAD).

The population background of each individual was estimated with peddy[81]. This revealed that the cohort was of predominantly European origin (Supplementary Table 1 and Supplementary Fig. 10).

Autozygosity was estimated using RohHunter, bcftools/roh or a sliding-window framework[82–84]. A small subset of samples was run

on all three tools, and this yielded comparable results for autozygosity. A threshold of 2% was used to assign patients to a high- or a low-autozygosity group[14] (Supplementary Fig. 11).

The variants identified in exome sequencing were assessed in accordance with the standards and guidelines of the ACMG for the interpretation of sequence variants[85]. At least two physicians or experts in molecular genetics participated in the assessment of the variants. Finally, all variants that were potentially disease-causing (pathogenicity class 3–5) and actionable secondary findings were reported to the respective patients.

Cases in which no diagnosis could be established in a known disease-associated gene were included in national and international studies for the discovery of novel disease etiologies for example, via the MatchMaker Exchange network[86,87]. Variants with a high likelihood of being disease-causing, for example, those with loss of function or high pathogenicity scores, or those that had arisen de novo, were shared through MatchMaker Exchange or a similar network in order to identify similar patients[88,89].

### Statistical analyses

All statistical analyses were conducted in R (version 4.2.2)[90]. Proportions were tested using a two-sided Fisher's exact test. The significance level was set to $\alpha = 0.05$, and P values were corrected via Bonferroni correction if necessary.

### Visualization of phenotype space using UMAP

First, data on known diseases and their clinical features were downloaded from the HPO website (https://hpo.jax.org/app/download/annotation, file: genes_to_phenotype.txt, downloaded on 10 April 2021). The disease data were merged with the data of the 1,577 individuals from TRANSLATE NAMSE by treating each disease–ID as one individual. Similarities in HPO terms between all pairs of individuals were then calculated using the R package ontologySimilarity (version 2.5). The similarities were then converted to a distance matrix and projected into a four-dimensional space using uniform manifold approximation and projection (UMAP). Subsequently, the first two dimensions of this projection were plotted using ggplot2 (version 3.3.4).

### Variants amenable to carrier screening

In cases with autosomal recessive inheritance, disease-causing variants in ClinVar were queried in January 2017 (beginning of the project) to take into account the state of knowledge available at the time of analysis. Variants were classified as amenable to carrier screening if they were classified as pathogenic or likely pathogenic in ClinVar or if they were predicted loss-of-function variants that were not predicted to escape nonsense-mediated messenger RNA decay. In compound-heterozygous inheritance, both variants were required to be (likely) pathogenic.

### Comparison of disease-associated genes reported in TRANSLATE NAMSE with those reported in other cohorts

In the German healthcare system, genetic testing of the more frequent rare disorders, for example, retinitis pigmentosa or hearing impairment, is performed using gene panels.

For a comparison with the cohort from the NIHR BioResource described in Turro et al.[20], all disease-associated genes were first ranked according to the frequency of submissions of pathogenic and likely pathogenic variants to ClinVar. Disorders caused by genes in the first quartile of the ClinVar gene distribution, such as *USH2A*, *ABCA4* and *BMPR2*, are more prevalent than phenotypes associated with genes in the fourth quartile. In addition, the year in which phenotype–gene associations had first been reported was determined to assess when a diagnosis could first have been established. The characteristics of the variants identified in the TRANSLATE NAMSE exome sequencing cohort were then compared with those identified in a cohort reported by Turro et al. in 2020.

Turro et al. subjected DNA from 9,802 individuals with a suspected rare disease to genome sequencing and reported pathogenic or likely pathogenic variants in 1,138 cases[20]. Around a quarter of these variants were assigned to genes with a high disease prevalence (Supplementary Fig. 4). In contrast, most disease-associated genes identified in the TRANSLATE NAMSE cohort were ultrarare, and more frequent diagnoses were underrepresented.

### Novel disease candidate genes

Sequence data from the unsolved cases were analyzed for variants in potential novel disease candidate genes. The following mandatory criteria for novel disease candidate genes were defined: (1) the gene had shown no previous robust association with any human phenotype; (2) no other clearly causative disease explanation was found; (3) the allele frequency of the respective variant was below the minor allele frequency cutoff or the variant was absent in controls; (4) inheritance was in accordance with the phenotype in the family and/or the variant co-segregated with the disease in multiple affected family members. As in the ClinGen approach and as suggested by others, characteristics, including gnomAD constraint metrics, inheritance and functional data, by which the level of evidence for the manually identified candidate genes could be assessed were defined[61,91,92] (Supplementary Table 3). An evidence score was then calculated, which could reach a maximum value of 8. Three of the nine criteria can only be applied to genes with an autosomal dominant mode of inheritance (de novo status and gnomAD constraint metrics), rendering the score less informative for autosomal recessive inheritance. For autosomal dominant inheritance, a score of 1–3 was ranked as medium evidence and a score of 4 and above as high evidence. For recessive inheritance, a score of 3 or above was ranked as high evidence and a score of below 3 was ranked as medium evidence. Genes first published as disease-associated during the course of TRANSLATE NAMSE were classified as novel DGG.

### Diagnostic yield prediction (YieldPred)

The TRANSLATE NAMSE exome sequencing cohort ($n = 1,577$) was randomly divided into a training set comprising 1,256 cases (399 solved, 32%) and a test set comprising 321 cases (99 solved, 31%). The binary status of a case (1, solved; 0, unsolved) was regressed on the 49 HPO-defined subcategories (cf. clinical and laboratory phenotype data) using LASSO for binary outcomes with the logit function as a link function (R package glmnet, version 4.1-4) and by controlling for age (adult/child), sex (male/female), sequencing laboratory and the use of the PEDIA workflow. Variable selection was applied on the 49 HPO-defined subcategories only. The model was fitted on the training set, and the penalty parameter was tuned via tenfold cross-validation. The resulting model was then applied to the test set, and its predictive performance was evaluated using the receiver operator characteristics curve.

We further validated the influence of the separate HPO terms on the model. Figure 5 shows the resulting coefficient plot and was checked for plausibility. We found a positive correlation between the number of HPO terms and the predicted probability on the complete TRANSLATE NAMSE exome sequencing cohort ($n = 1,577$; Supplementary Fig. 6). Since the approach of HPO-defined subcategories ensures that multiple lower-order terms are only counted once, this finding indicates that a monogenic cause and diagnosis via exome sequencing is more likely if a patient exhibits a diverse set of clinical features. Furthermore, we investigated the discriminatory power of all 1,649 unique HPO terms that were annotated in the TRANSLATE NAMSE cohort. Considering each HPO term separately to discriminate between solved and unsolved patients led to an average AUC of 0.5 (s.d. 0.003), that is, no discriminatory power. The maximum achieved AUC of a single HPO term, namely HP:0001263 (global developmental delay), was 0.58. As a sensitivity analysis, we then fitted a logistic regression on the complete TNAMSE cohort with the top five HPO

terms, namely HP:0001263 (global developmental delay), HP:0000252 (microcephaly), HP:0001252 (hypotonia), HP:0001250 (seizure) and HP:0001251 (ataxia), and achieved an AUC of 0.64 (95% CI 0.61–0.67). On the complete TNAMSE set (that is, training and test set combined) our YieldPred model yielded an AUC of 0.72 (95% CI 0.69–0.74). In summary, there are some HPO terms that have higher discriminatory power than the majority of the HPO terms. However, the signal of YieldPred is additionally driven by the combination of multiple phenotypic features that are present in a patient.

To increase the portability and applicability of the Lasso model, two additional external and independent cohorts were included. This first external cohort (*n* = 753, 545 solved, 72%; Supplementary Table 5) was recruited by the Technical University of Munich, and all individuals consented in the scientific use of their phenotype and genotype data. As a second external cohort, we used the NIHR BioResource cohort described by Turro et al. (*n* = 5,510, 1,059 solved, 19%). The Lasso model was then retrained on cases of all three cohorts and 20% of the cases of each cohort were kept as hold-out test set. The AUCs of the final model ranged from 0.64 for the TRANSLATE NAMSE cases of the test set and 0.65 for the Munich cases of the test set to 0.71 for the cases of the test set from the cohort of Turro et al. (Supplementary Fig. 5). The final model was provided as the tool YieldPred as a web service, where users can specify the age, sex and assigned HPO terms of their patient, while the remaining confounders are estimated via the mean confounder values of the training cohort.

### PEDIA analysis

PEDIA integrated the facial image and clinical feature analysis with exome data analysis[41]. For each patient, a frontal facial image, clinical features encoded in HPO terminology, and exome sequencing data were available for analysis.

The PEDIA approach was used, in which the facial image analysis was analyzed by GestaltMatcher[40]. GestaltMatcher was trained on 6,354 frontal images with 204 different disorders to learn the respective facial dysmorphic features, and it further encoded each image into a 512-dimensional facial phenotype descriptor. The model ensembles and test-time augmentation were later used to generate 12 512-dimensional facial phenotype descriptors for each image[93]. The similarity between two patients can be quantified by averaging 12 cosine distances of the facial phenotype descriptors. For each test image, a list of similarity scores for 816 disease-causing genes were obtained. To convert HPO terms of individual patients into feature scores for each gene, the CADA approach was used[69]. For the exome data, each variant was annotated with a version 1.6 CADD score[42]. After filtering out the common variants, the highest CADD score for each gene was taken.

In this analysis, benchmarking was performed on two cohorts: the PEDIA subcohort and the validation cohort. The PEDIA subcohort consisted of a subset of 224 of the 1,577 exome sequencing patients (194 pediatric, 30 adult). Of these, 94 had a molecular genetic diagnosis (86 pediatric, 8 adult). After the end of the 3-year TRANSLATE NAMSE recruitment period, a further 149 patients were enrolled and used as a validation cohort. In the validation cohort, 69 out of 149 patients were solved cases. All facial images analyzed in the present study can be accessed in GestaltMatcher Database (https://db.gestaltmatcher.org/) by the GMDB ID in Supplementary Tables 1 and 6. For each patient, each gene had a GestaltMatcher score, a CADA score and a CADD score. These three scores were the input of the PEDIA approach. The output for each patient was a list of genes, and each gene had a PEDIA score. The genes were then prioritized by ranking the PEDIA scores in descending order. To benchmark the performance, top-*k* accuracy was used, as calculated by the percentage of the patients with the disease-causing gene ranked in the top-*k* position. Finally, the top-1 to top-100 accuracies of the two cohorts (the PEDIA subcohort of the exome sequencing cohort and validation cohort) were reported.

### Reporting summary

Further information on research design is available in the Nature Portfolio Reporting Summary linked to this article.

## Data availability

The corresponding author agrees to fulfill any requests for materials not included in the article, subject to verification that the request adheres to the consent provided by the research participants. Patient-related data not included in the article may be subject to patient confidentiality. Raw sequencing data were not consented for sharing, except for the PEDIA subset, which is available upon request. Reported alleles and their clinical interpretation have been deposited in ClinVar using the following submitters: Institute for Genomic Statistics and Bioinformatics (University Hospital Bonn) (https://www.ncbi.nlm.nih.gov/clinvar/submitters/507028/, https://www.ncbi.nlm.nih.gov/clinvar/submitters/508040/); Institute of Human Genetics, Klinikum rechts der Isar (Technical University Munich) (https://www.ncbi.nlm.nih.gov/clinvar/submitters/500240/); Institute for Medical Genetics and Human Genetics (Charité – Universitätsmedizin Berlin) (https://www.ncbi.nlm.nih.gov/clinvar/submitters/505735/); Institute of Medical Genetics and Applied Genomics (University Hospital Tübingen) (https://www.ncbi.nlm.nih.gov/clinvar/submitters/506385/); and Genomics Facility (Ludwig-Maximilians-Universität München) (https://www.ncbi.nlm.nih.gov/clinvar/submitters/507363/).

## Code availability

The study's landing page (https://www.translate-namse.de) redirects to a web service for the prediction of the diagnostic yield and the code repository at GitHub (https://github.com/Ax-Sch/TNAMSE_geno_pheno). Code is also available via Zenodo at https://doi.org/10.5281/zenodo.10964188 (ref. 94). All source codes are available under a creative commons license.

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

## Acknowledgements

We thank all patients and families from TRANSLATE NAMSE and NIHR BioResource for their cooperation. We thank C. Schmael for proofreading of the manuscript. M.D., H.L.S. and M.A.M. are participants in the BIH Charité (Digital/Junior) Clinician Scientist Program, which is funded by Charité – Universitätsmedizin Berlin and the Berlin Institute of Health (BIH). F. Boschann is a participant in the Clinician Scientist Program (CS4RARE) funded by the Alliance4Rare and associated to the BIH Charité Clinician Scientist Program. A.S. was supported by the BONFOR program of the Medical Faculty, University of Bonn (O-149.0134). M.A.L.-K. received funding from DFG (CRC237 369799452/B21 and CRC237 369799452/A11). C. Schlein received funding from DFG (SCHL2276/2-1; 450149205-TRR333/1). E.T. was funded by NIH awards R01HL161365 and R03HD111492.

## Author contributions

Study conceptualization and design: N.E., T. Haack, P.M.K. and M.W. Sample and data acquisition: R.A.J., L.A., A.A., S.B.-W., M. Begemann, T. Bender, R. Berner, S.B., R. Berutti, M. Bettendorf, R.C.B., A. Bevot, I.B., F. Boschann, F. Brand, F. Brinkmann, M. Brugger, T. Brunet, P.B., T. Bäumer, A. Bösch, C.B., S.C.-G., D.C., K.C., M.D., G.D., N.D.D., N.E., M.E., H.E., H.G., E.G., C.G., L.G., M.G.-E., M.G., K.G., A.G.-K., T. Haack, B.H., A.-M.H., F.H., A. Heimbach, A. Heinen, Y.H., M.H., J. Hentschel, T. Herget, R. Herzog, K.H., R. Hirtz, J. Hoefele, S. Hoffjan, G.F.H., S. Holzhauer, D.H., I.H., A.M.K., F.J.K., N.K., T. Kallinich, P.K., V.K., L.T.K., C. Kehrer, M. Kehrer, C. Kiewart, O.K., C. Klein, T. Klockgether, A. Knaus, C. Knopp, X.K., U. Kornak, U. Kotzaeridou, R.K., F.K., P.M.K., M. Kreiss, M. Krenn, T. Kretschmer, H.K., C. Kubisch, A. Kuechler, S.K., I.K., J.K., M.A.L.-K., J.R.L., H.L., J.L., K.L., J.M., C.M., E. Mangold, E. Manka, N.M., K.M., T.M., M.A.M., N.M.-S., A.M.M., S. Mundlos, A.C.M., M. Munteanu, M. Mücke, W.M.-F., A.M., J.N., H.H.P.N., M.N., K.O., P.M.O., C.-E.O., J.T.P., M.P., C.P., S.P., K.P., U.R., K.M.R., O.R., F.R., A.R., M.R., T.R., C.R., C.P.S., U.A.S., E. Schlapakow, C. Schlein, A.S., C. Schramm, E. Schröck, S. Schröder, M. Schuelke, C. Schuetz, E.M.C.S., S. Schwartzmann, V.S.S., L.J.S., H.L.S., S. Sivalingam, M. Spielmann, I.S., C. Stieber, C. Stoltenburg, V.S., T.S., M. Sturm, A.-A.S., P.T., A.T., E.T., K.U., M.W., H.W., C.W.-N., N.W., C.W., D.S.W., P.W., M.Z., L.Z. and D.Ö. Analysis and interpretation: M.A.B., L.A., T. Brunet, M.D., G.D., N.E., H.E., K.G., T. Haack, M.H., T.-C.H., P.I., H. Klinkhammer, A. Knaus, P.M.K., S. Moosa, A.S., S. Sivalingam, E.T. and M.W. Manuscript writing: T. Brunet, M.D., N.E., H.E., K.G., T. Haack, T.-C.H., H. Klinkhammer, P.M.K., A.S. and M.W. Coordination and funding acquisition: N.E., T. Haack, P.M.K., H. Krude, T.M., S. Mundlos, M.N., O.R. and M.W.

## Competing interests

V.S.S. has received consultant fees from Novartis, Chugai, AbbVie, Celgene, Sanofi, Lilly, Hexal, Pfizer, Amgen, BMS, Roche, Gilead, Medac, Boehringer-Ingelheim and Alexion and speaker's bureau fees from AbbVie, Novartis, BMS, Chugai, Celgene, Medac, Sanofi, Lilly, Hexal, Pfizer, Janssen, Roche, Schire, Onkowissen, Royal College London, Boehringer-Ingelheim and UCB Fresenius. M.G.-E. has received research support from the German Ministry of Education and Research (BMBF) within the European Joint Program for Rare Diseases (EJP-RD) 2021 Transnational Call for Rare Disease Research Projects (funding number 01GM2110), from the National Ataxia Foundation (NAF) and from Ataxia UK and received consulting fees from Healthcare Manufaktur, Germany, all unrelated to this study. All other authors declare no competing interests.

## Additional information

**Correspondence and requests for materials** should be addressed to Peter M. Krawitz.

# natureportfolio

# Reporting Summary

## Statistics

For all statistical analyses, confirm that the following items are present in the figure legend, table legend, main text, or Methods section.

| n/a | Confirmed | |
|---|---|---|
| ☐ | ☒ | The exact sample size (*n*) for each experimental group/condition, given as a discrete number and unit of measurement |
| ☐ | ☒ | A statement on whether measurements were taken from distinct samples or whether the same sample was measured repeatedly |
| ☐ | ☒ | The statistical test(s) used AND whether they are one- or two-sided<br>*Only common tests should be described solely by name; describe more complex techniques in the Methods section.* |
| ☐ | ☒ | A description of all covariates tested |
| ☐ | ☒ | A description of any assumptions or corrections, such as tests of normality and adjustment for multiple comparisons |
| ☐ | ☒ | A full description of the statistical parameters including central tendency (e.g. means) or other basic estimates (e.g. regression coefficient) AND variation (e.g. standard deviation) or associated estimates of uncertainty (e.g. confidence intervals) |
| ☐ | ☒ | For null hypothesis testing, the test statistic (e.g. *F*, *t*, *r*) with confidence intervals, effect sizes, degrees of freedom and *P* value noted<br>*Give P values as exact values whenever suitable.* |
| ☐ | ☒ | For Bayesian analysis, information on the choice of priors and Markov chain Monte Carlo settings |
| ☐ | ☒ | For hierarchical and complex designs, identification of the appropriate level for tests and full reporting of outcomes |
| ☐ | ☒ | Estimates of effect sizes (e.g. Cohen's *d*, Pearson's *r*), indicating how they were calculated |

*Our web collection on statistics for biologists contains articles on many of the points above.*

## Software and code

Policy information about availability of computer code

| Data collection | For individuals additionally participating in the PEDIA study, scores of the analysis of the portrait images by artificial intelligence (PEDIA v3) were collected via the PEDIA web service (https://www.pedia-study.org/). |
|---|---|
| Data analysis | All analyses that have been conducted on the data are described in the main manuscript and the supplement. The regarding code can be found in the publicly available github respository cited in the manuscript (https://github.com/Ax-Sch/TNAMSE_geno_pheno). The analyses were conducted in the statistics software R (version 4.2.2).<br>Raw exome data were analysed with BWA v0.7.11 through to BWA-Mem v0.7.17, HaplotypeCaller (v3.7, v3.8 or v4.1), Freebayes (v1.2.0), SAMtools v.0.1.7, ExomeDepth v1.1.10 , ClinCNV v.1.16.1, ngs-bits v2019_09, VEP v96 and Jannovar v0.24. |

For manuscripts utilizing custom algorithms or software that are central to the research but not yet described in published literature, software must be made available to editors and reviewers. We strongly encourage code deposition in a community repository (e.g. GitHub). See the Nature Portfolio guidelines for submitting code & software for further information.

## Data

Policy information about availability of data

All manuscripts must include a data availability statement. This statement should provide the following information, where applicable:
- Accession codes, unique identifiers, or web links for publicly available datasets
- A description of any restrictions on data availability
- For clinical datasets or third party data, please ensure that the statement adheres to our policy

The corresponding author agrees to fulfill any requests for materials not included in the extended data or Supplementary Material, subject to verification that the request adheres to the consent provided by the research participants. Patient-related data not included in the article may be subject to patient confidentiality.  Raw

sequencing data was not consented for sharing, except for the PEDIA subset which is available upon request. Reported alleles and their clinical interpretation have been deposited in ClinVar using the following submitters:

Institute for Genomic Statistics and Bioinformatics (University Hospital Bonn):
https://www.ncbi.nlm.nih.gov/clinvar/submitters/507028/,
https://www.ncbi.nlm.nih.gov/clinvar/submitters/508040/
Institute of Human Genetics, Klinikum rechts der Isar (Technical University Munich):
https://www.ncbi.nlm.nih.gov/clinvar/submitters/500240/
Institute for Medical Genetics and Human Genetics (Charité- Universitätsmedizin Berlin):
https://www.ncbi.nlm.nih.gov/clinvar/submitters/505735/,
Institute of Medical Genetics and Applied Genomics (University Hospital Tübingen):
https://www.ncbi.nlm.nih.gov/clinvar/submitters/506385/.
Genomics Facility (Ludwig-Maximilians-Universität München):
https://www.ncbi.nlm.nih.gov/clinvar/submitters/507363/

# Field-specific reporting

Please select the one below that is the best fit for your research. If you are not sure, read the appropriate sections before making your selection.

☒ Life sciences          ☐ Behavioural & social sciences          ☐ Ecological, evolutionary & environmental sciences

For a reference copy of the document with all sections, see nature.com/documents/nr-reporting-summary-flat.pdf

# Life sciences study design

All studies must disclose on these points even when the disclosure is negative.

| | |
|---|---|
| Sample size | In total, 5652 individuals with a suspected rare disorder were enrolled in TRANSLATE-NAMSE by centers for rare diseases at ten German university hospitals over a period of three years (2018-2020), i.e. over the complete duration of the study. A cohort of 1577 patients underwent exome sequencing with recommendation of multidisciplinary teams. In this manuscript we report results for those 1577 patients of which 211 patients additionally consented to the analysis of their potraits by an artificial intelligence tool (PEDIA). For the evaluation of the exome sequencing cohort, no sample size calculation was performed beforehand, but all eligible samples were included in the study and there was no option to further increase the sample size. The achieved sample size was considered sufficient for descriptive statistics. For the final predictive model (YieldPred), additional cohorts were included to increase the sample size (NIHR BioResource, n=5,510 and external validation cohort, n=753). |
| Data exclusions | No data were excluded. |
| Replication | All findings of the analysis can be reproduced by the provided code (https://github.com/Ax-Sch/TNAMSE_geno_pheno). We validated the findings of the LASSO model on an external patient cohort (shown in the supplement) and the NIHR BioResource. |
| Randomization | For the LASSO analysis the patients were randomly assigned to training (80%) and test (20%) data sets. All other analyses were conducted on the complete TRANSLATE-NAMSE exome sequencing cohort. |
| Blinding | Splitting data into training and test sets is equivalent to blinding data for the machine. |

# Reporting for specific materials, systems and methods

We require information from authors about some types of materials, experimental systems and methods used in many studies. Here, indicate whether each material, system or method listed is relevant to your study. If you are not sure if a list item applies to your research, read the appropriate section before selecting a response.

### Materials & experimental systems

| n/a | Involved in the study |
|---|---|
| ☒ | ☐ Antibodies |
| ☒ | ☐ Eukaryotic cell lines |
| ☒ | ☐ Palaeontology and archaeology |
| ☒ | ☐ Animals and other organisms |
| ☐ | ☒ Human research participants |
| ☒ | ☐ Clinical data |
| ☒ | ☐ Dual use research of concern |

### Methods

| n/a | Involved in the study |
|---|---|
| ☒ | ☐ ChIP-seq |
| ☒ | ☐ Flow cytometry |
| ☒ | ☐ MRI-based neuroimaging |

# Human research participants

Policy information about <u>studies involving human research participants</u>

| Population characteristics | Male and female as well as pediatric and adult participants were analyzed in the study. All patients underwent exome sequencing while a subset of patients additionally consented to image analysis of their potraits by artificial intelligence (PEDIA). Age (below or above 18 years), sex, the sequencing laboratory as well as the use of the PEDIA workflow could confound the diagnostic yield and the data were therefore included as confounders in the LASSO analysis. |
|---|---|
| Recruitment | Participants were enrolled in the TRANSLATE-NAMSE study at ten centers for rare diseases at German university hospitals and recommended to undergo exome sequencing by multidisciplinary teams. |
| Ethics oversight | This study is governed by the approval of the following Institutional Review Boards: Charité–Universitätsmedizin Berlin, Germany (EA2/140/17); UKB Universitätsklinikum Bonn, Germany (Lfd.Nr.386/17); Universitätsklinikum Essen, University Duisburg-Essen, Germany (17-7774-BO); Universitätsklinikum Heidelberg, Germany (S-499/2017); Universitätsklinikum Tübingen, Germany (643/2017BO1); Universität zu Lübeck, Germany (17-272); Ludwig-Maximilians-Universität München, Germany (17-640); Ärztekammer Hamburg, Germany (MC-316/17); Technische Universität Dresden, Germany (AK 464122017). All patients or their legal guardians provided written informed consent prior to inclusion. |

Note that full information on the approval of the study protocol must also be provided in the manuscript.

