## [Peer Review File · Nature Genetics]

Peer Review Information

Manuscript Title: Next-generation phenotyping integrated in a national framework for patients with ultra-rare disorders improves genetic diagnostics and yields new molecular findings

Corresponding author name(s): Professor Peter Krawitz

Reviewer Comments & Decisions:

Decision Letter, initial version:
--

7th September 2023

Dear Peter,

Your Article "Next-generation phenotyping integrated in a national framework for patients with ultra-rare disorders improves genetic diagnostics and yields new molecular findings" has been seen by three referees. You will see from their comments below that, while they find your work of interest, they have raised several important points. We are interested in the possibility of publishing your study in Nature Genetics, but we would like to consider your response to these points in the form of a revised manuscript before we make a final decision on publication.

To guide the scope of the revisions, the editors discuss the referee reports in detail within the team, including with the chief editor, with a view to identifying key priorities that should be addressed in revision, and sometimes overruling referee requests that are deemed beyond the scope of the current study. In this case, we ask that you address all queries related to the analysis pipeline, clarifying the presentation as needed, and perform additional benchmarking to evaluate the performance of PEDIA relative to other widely used clinical diagnostic pipelines. We hope you will find this prioritized set of referee points to be useful when revising your study. Please do not hesitate to get in touch if you would like to discuss these issues further.

We therefore invite you to revise your manuscript taking into account all reviewer and editor comments. Please highlight all changes in the manuscript text file. At this stage, we will need you to upload a copy of the manuscript in MS Word .docx or similar editable format.

*2) If you have not done so already, please begin to revise your manuscript so that it conforms to our Article format instructions, available here.
Refer also to any guidelines provided in this letter.

Please be aware of our guidelines on digital image standards.

[redacted]

We hope to receive your revised manuscript within 8-12 weeks. If you cannot send it within this time, please let us know.

Nature Genetics is committed to improving transparency in authorship. As part of our efforts in this direction, we are now requesting that all authors identified as 'corresponding author' on published papers create and link their Open Researcher and Contributor Identifier (ORCID) with their account on the Manuscript Tracking System (MTS), prior to acceptance. ORCID helps the scientific community achieve unambiguous attribution of all scholarly contributions. You can create and link your ORCID from the home page of the MTS by clicking on 'Modify my Springer Nature account'. For more information, please visit www.springernature.com/orcid.

Sincerely,
Kyle

Kyle Vogan, PhD

Senior Editor
Nature Genetics
<https://orcid.org/0000-0001-9565-9665>

Referee expertise:

Referee #1: Clinical genomics, rare diseases, bioinformatics

Referee #2: Clinical genomics, rare diseases, pediatrics

Referee #3: Clinical genomics, rare diseases, pediatrics

Reviewers' Comments:

Reviewer #1:
Remarks to the Author:

Schmidt et al present interesting results from the German TRANSLATE NAMSE project for ultra-rare disease. However, the paper lacks a bit of focus on what the main message is. Is it the diagnostic yield and lessons learned on 1,577 patients in TRANSLATE NAMSE, the ability to predict which patients will receive a diagnosis from WES (YieldPred), or the power of AI to improve diagnosis and discovery? From the title, it seems the last message is the key one and is certainly the most interesting one.

For the first message, compared to other recent studies, the cohort size is not that large, but the focus on ultra-rare diseases is a novel angle. It would be good to better define what they mean by ultra-rare diseases from the beginning though.

The YieldPred tool is intriguing and of potential use to all clinicians performing genetic testing of rare disease patients. The training sample size does seem small and quite focussed on neurological disorders and it would be good to discuss more the content of the independent validation cohort and whether they expect YieldPred to perform well on rare disease areas not covered by TRANSLATE NAMSE.

The most interesting aspect is the application of AI image analysis approaches in PEDIA. A performance of 82% diagnoses identified in the top 10 ranked candidates is reported, which is an improvement over their other approaches, including an HPO-based method (CADA). However, lots of other tools are routinely used in clinical diagnostics such as Exomiser, xRare, AMELIE and LIRICAL. The former in particular is used in the UK's Genomic Medicine Service and 100,000 Genomes Project and 88% of diagnoses were reported in the top 5 candidates. Some comparison to tools developed by other groups is needed before the performance of PEDIA can really be judged.

Minor comments
Add percentages to diagnostic yield numbers in the abstract

Reviewer #2:

Remarks to the Author:

Authors describe a large German national exome sequencing project, TRANSLATE NAMSE, involving 10 hospital sites that recruited, sequenced, analyzed, and diagnosed patients/participants over 3 years. Diagnosis rates are generally on par with other large studies of this type (31.6%) and project contributes to 57 novel disease gene relationship discoveries (~40% still candidates). Overall, this is a nice summary of the program and involves using the data to develop some new tools (though how they are applied to improve diagnosis rates in this cohort was not entirely clear). Overall, this is a well-done study that will be of high interest to the human genomics community.

Major concerns:

The ClinVar ultra-rare analysis is interesting, but I think it was done by number of variants per gene rather than number of submissions per gene (though I am not entirely sure from the methods). It's not been established that disease prevalence is correlated with number of pathogenic variants, as this does not take into account the prevalence of these variants in the general population (with inheritance mode). Additionally, some of the genes with the most pathogenic variants may be from population screening of genes on the secondary findings list (who may not all develop disease) rather than from testing on patients with rare disease. This approach of counting unique variants will make the gain of function diseases that have a smaller repertoire of pathogenic variants appear more rare – and also a lab may submit one variant classification but have seen the variant multiple times. On the other hand, the decade of gene-disease relationship discovery should be more robust. It's an interesting if flawed approach that could be improved but not completely corrected. The approach is perhaps better applied as a comparison method, like they also use it here to the 2020 Turro et al. cohort (with a 16% diagnosis rate from genome sequencing). The authors attribute this result of seeing rarer conditions in this Schmidt et al. cohort to the benefit of MDT involvement rather than other differences between the cohort (prior testing, family structure, etc.). While there are many ultra-rare conditions identified in this cohort, one might expect to also see more common conditions, so the lack of more common conditions raises questions on why they are not present.

"Functional assays": This section was disorganized. Consider handling functional assays for follow-up of VUS as a paragraph (which were very successful here) and untargeted functional assays (RNAseq which did not add diagnoses) as a separate paragraph. Also, 5/18 cases are mentioned but 21 are mentioned suggesting some cases had multiple functional studies. Would also be helpful to mention the tissue that was used for each functional assay.

YieldPred is a statistical framework for predicting the likelihood of a molecular diagnosis based on clinical features. It is not clear how this tool is contributing to the goals of this project though this dataset was used for training and testing the method. It is unclear how robust the tool is to HPO term entry.

The validity of the model was not well established here. Do predictions correlate with number of HPO terms? Are a few HPO terms driving the signal?

The utility of the YieldPred in this study is not well described. Was YieldPred used to identify cases for reanalysis, additional functional studies (RNAseq), something else?

Related, I did not understand this sentence in the discussion: "Low scores in YieldPred despite a high likelihood of a monogenic cause, e.g. a strong family history, could justify the selection of an even more comprehensive test, such as long-read genome sequencing."

"Diagnoses with causal therapeutic implications" describes 5 patients (0.3% cohort, 1% diagnosed cohort) with specific targeted therapy though not a cure for all (2 were ketogenic diet). Changes in medical management are not described and may have not been systematically assessed but seems warranted to mention that diagnoses resulted in additional management changes, if that is the case.

The PEDIA tool was also described more in terms of how well it works rather than being used to diagnose additional cases in this cohort, through one case involving KANSL was detailed.

Minor concerns:

"Diagnostic yield of ES": please clarify if this referred to only pathogenic and likely pathogenic variants or also included high interest variants of uncertain significance to be clear how the diagnostic rate is calculated.

- These VUS are also called variants of "unknown significance" in the text which is not the typical term used in the field.

- "guidelines of the American College of Medical Genetics and Genomics (ACMG) and the Association for Molecular Pathologists (AMP) for classification of sequence variants"

Secondary findings are described in the supplement but could warrant brief mention in the main manuscript.

While homozygous variants were more likely to be the diagnosis in cases with consanguinity, it is worth noting that 25% of these cases had another mode of inheritance, highlighting the importance of looking more broadly.

The reference numbering is off in the manuscript and needs to be addressed.

Data availability: It is not clear from the manuscript or the referenced study website if there is an ability for researchers to obtain access to the CRAMs and vcf files from the project and how this access is governed.

Reviewer #3:

Remarks to the Author:

Thank you for the opportunity to review this manuscript, which presents the outcomes of a national program to improve rare disease diagnosis in Germany, which combined expert patient evaluation with ES. The study piloted the use of machine learning and AI-based approaches to augment diagnosis and has made a significant contribution to the delineation of novel gene-disease associations, as well as impacting policy in Germany in terms of increasing access to funded genomic testing.

This study is set up using well established paradigms, and outcomes of several such national programs have been published, notably Care4Rare Canada and the DDD study in the UK, which have each been operational for over 10 years, and have reported outcomes on ~1,800 and ~13,000 probands respectively.

The diagnostic yield reported (including breakdowns by inheritance pattern, contribution of dual diagnoses and ultra-rare conditions) and the rate of novel gene-discovery are consistent with previously published studies.

It is unclear how the 1,577 patients reported here were selected from the bigger cohort of 5,652 participants.

The manuscript would benefit from more detail on how variants were prioritized for curation and reporting apart from having experienced clinicians/scientists involved after variant annotation. Typically, multiple tools, parameters and analysis strategies are used by diagnostic laboratories to ensure appropriate variants are reported and curated. For example, what proportion of patients were analysed as singletons vs trios? Were virtual panels applied based on phenotype? Which variant types exactly were included in analysis? If CNVs were systematically searched for, then what tools were used? which variant types were not systematically searched for? e.g. structural variants, UPD, mosaic?

The additional advantage of using PEDIA needs to be benchmarked more appropriately. I am not aware of any diagnostic laboratory that solely uses CADD score or other in-silico tools to prioritise variants. There are multiple tools designed to prioritise variants based on a combination of HPO terms and variant properties (e.g. Exomiser) and a selection of these would be a more appropriate comparator.

While I agree that the MDT approach to patient selection is well recognised to increase diagnostic yield and I am not surprised it enriched the cohort for ultra-rare diagnoses, it is also important to recognise that over time, particularly after the establishment phase, it can act as a barrier to access and to scalability. Given the large body of literature already published about the high diagnostic yield in specific patient groups (e.g. severe intellectual disability, infantile epilepsy), a guideline-based approach may be more appropriate with MDT-based patient selection at academic centres reserved for more unusual clinical presentations and/or after an initial exome test is uninformative.

Author Rebuttal to Initial comments

Reviewer #1: For the first message, compared to other recent studies, the cohort size is not that large, but the focus on ultra-rare diseases is a novel angle. It would be good to better define what they mean by ultra-rare diseases from the beginning though.

Response: We agree with the reviewer that the category of ultra-rare diseases requires a definition in our manuscript. Wakap, *et al.* estimated that more than 80% of the rare diseases

have a prevalence below one in a million. However, at these low frequencies the proportion of undiagnosed patients is probably very high, and this is most likely also true for the novel disorders that we could establish in our work. In order to foster clinical trials and the development of orphan medicinal products the EU, therefore, used a much higher threshold for ultra-rare, which is “[...] a severe, debilitating and often life-threatening disease affecting no more than one person in 50,000 in the Union [...]” Regulation (eu) no 536/2014, (Sobrido et al. 2019).

In the introduction of the manuscript, we now refer to this definition of the ultra-rare disorders by prevalence, and also point out the long tail distribution:

Rare diseases thus represent a substantial global health burden. However, only a minority of suspected rare disease patients receive both a definite clinical diagnosis and a confirmatory molecular test result^{2,3}. This concerns in particular the subset of patients with ultra-rare disorders that are defined in the European Union by affecting no more than one person in 50,000 and that follow a long tail distribution with respect to their frequency (Regulation (eu) no 536/2014). It is estimated that roughly 80% of the more than 5000 rare diseases have a prevalence below one in a million¹.

Reviewer #1: The YieldPred tool is intriguing and of potential use to all clinicians performing genetic testing of rare disease patients. The training sample size does seem small and quite focused on neurological disorders and it would be good to discuss more the content of the independent validation cohort and whether they expect YieldPred to perform well on rare disease areas not covered by TRANSLATE NAMSE.

Response: We agree with the reviewer that our cohort and therefore also our training sample size is relatively small in comparison to other data sets. We also acknowledge that there is a higher proportion of neurological disorders in the TRANSLATE NAMSE cohort compared to our validation cohort and other cohorts. We therefore contacted Ernest Turro for some additional analyses on the NIH BioResource cohort that he and his colleagues described in *Nature* in 2020: *Whole-genome sequencing of patients with rare diseases in a national health system*.

We find the comparison intriguing because it also reveals some differences in the healthcare systems that we will discuss in the following and added some of the additional results to the supplement.

First, we visualized the phenotypic differences of the three cohorts by projecting all cases that were annotated with HPO terms on the same map of OMIM and Orphanet disorders. Please note, that this results in a new mapping with slightly changed proportions in comparison to the old *Figure 1 C* of the main manuscript.

Figure 1: The phenotypic similarity of all cases was visualized based on their clinical features with the tool kit ontology similarity. The proportion of the mapping changes with the sets of cases that were included. We now added a comparison of the TRANSLATE NAMSE cohort with our validation cohort and the cohort from Turro, *et al.* to the supplement and replace the old figure 1 C in our main manuscript by the new mapping.

In the original work of Turro, *et al.*, all cases were assigned to one of 15 disease domains that differed from our six disease groups. For better comparison we now use the same color code which is based on the leading clinical feature which is encoded by a higher order HPO term. This encoding was already introduced in the old Supplementary Material and is also related to the parameters used in the Lasso analysis. In comparison we now see the differences in the distribution over the phenotypic map indicating a different composition of the cohorts. The TRANSLATE NAMSE dataset comprises individuals with a relatively even distribution across the whole disease landscape. In contrast, the cohort published by Turro, *et al.* has more individuals with pulmonary arterial hypertension, immune disorders and bleeding, thrombotic and platelet disorders, which can most likely be explained by differences in the healthcare system. In Germany most of the cases with e.g. pathogenic variants in *BMPT2* are diagnosed by panel sequencing and did not participate in TRANSLATE NAMSE.

These phenotypic differences also most likely explain the drop in AUC from 0.67 for the TRANSLATE NAMSE holdout set to AUC=0.58 for our validation cohort and to 0.58 for the cohort described by Turro, *et al.* if the original Lasso model is applied to these cohorts. We can therefore confirm that the portability of the Lasso model that was only trained on the TRANSLATE NAMSE cohort is limited for cases from phenotypically different cohorts.

A, TNAMSE, taken from figure 1

B, TNAMSE, colored by HPO-terms

C, internal validation cohort

D, cohort from Turro, et al.

Supplementary Figure 3: Phenotypic comparison of three cohorts. TNAMSE is depicted in A and B. The validation cohort is depicted in C. The cohort from Turro, et al. is depicted in D. The color code in A is as in Figure 1 of the main manuscript, where each case is assigned to one disease group. In B, C, and D, each case is assigned to a single higher order HPO term that was assessed by a clinician as leading clinical feature. The clusters are differently densely populated and indicate phenotypic differences in the three cohorts.

In order to increase the applicability, we trained a new Lasso model on cases of all three cohorts and kept 20% of the cases of each cohort as hold-out test set. The AUCs for the new model on the test set of TRANSLATE NAMSE is 0.64, which is comparable to 0.67 of the old model. For

the test sets of the validation cohort, and the cohort of Turro, *et al.*, however, the AUCs are greatly improved to 0.65 and 0.71 indicating better generalizability of the model. While this increase was expected for the cohort of Turro, *et al.*, which makes up the majority of the training cohort, the increase in the performance on the validation cohort and the relatively stable performance on the TRANSLATE NAMSE cohort suggest that the model is not overfitting to the cohort of Turro, *et al.* We appreciate very much that the reviewer sees clinical utility in YieldPred for expectation management beyond the simply identifying phenotypic features that showed a positive or negative effect in our cohort in terms of the probability of establishing a molecular diagnosis. We, therefore, made this improved model already available as a webservice and added the discussion to the Supplementary material. In future research, we would like to investigate in more detail whether the probability of establishing a molecular diagnosis with a certain test based on phenotypic features is superior to an estimate that is based on a disease group or a primary phenotype (Retterer et al. 2016). For the additional analyses we would also like to add Ernest Turro as a coauthor.

Supplementary Figure 5: Receiver operator characteristics (ROC) for the original Lasso model that was only trained and tested on cases of the TRANSLATE NAMSE cohort (A) and the new Lasso model that was trained and tested on cases of all three cohorts (B-D). For testing, 20% of cases were used as hold-out sets. For the new model the AUC for the TRANSLATE NAMSE test set drops slightly from 0.67 to 0.64. However, the generalization is markedly improved, achieving an AUC of 0.65 on the test set of the validation cohort and 0.71 on the test set of the NIHR BioResource cohort described by Turro, et al.

Reviewer #1: The most interesting aspect is the application of AI image analysis approaches in PEDIA. A performance of 82% diagnoses identified in the top 10 ranked candidates is reported, which is an improvement over their other approaches, including an HPO-based method (CADA). However, lots of other tools are routinely used in clinical diagnostics such as Exomiser, xRare, AMELIE and LIRICAL. The former in particular is used in the UK's Genomic Medicine Service and 100,000 Genomes Project and 88% of diagnoses were reported in the top 5 candidates.

Some comparison to tools developed by other groups is needed before the performance of PEDIA can really be judged.

Response: We agree with the reviewer that there are many possible combinations of how the scores of the image analysis can be combined with results of other prioritization tools for clinical features such as CADA and molecular pathogenicity scores such as CADD. Since the PEDIA approach is modular, we therefore analyzed whether image analysis with GestaltMatcher can also improve the performance of the suggested prioritization approaches Exomiser, Xrare, Lirical, and Amelie. First, we trained a support vector machine (SVM) for each of the tools on the cohort data of the original publication so that a PEDIA score could be computed for the combinations of scores (Hsieh et al. 2019). Then, we tested the top-k-accuracy rates on the TNAMSE cohort, as well as on the validation set.

The following Figures show the performance comparison between using each of the four methods alone and in combination with GestaltMatcher. Integrating GestaltMatcher scores with each method improved the Top-1 accuracy by 4 to 38 percentage points and the Top-10 accuracy by 6 to 17 percentage points. Similar results also are shown on the validation cohort (Figure 2). Therefore, we conclude that the GestaltMatcher can be integrated not only with CADA and CADD scores but also the other variant prioritization approaches. We added these analyses to the Supplementary Material and cover this shortly in the main results. We also refer the interested audience to a Current Protocol about how to the open source tool GestaltMatcher via a REST API (Hsieh, Lesmann, and Krawitz 2023).

The PEDIA approach is highly modular and the GestaltMatcher score for image analysis can also be combined with other prioritization tools such as Exomiser⁴², Xrare⁴³, Lirical⁴⁴, or Amelie⁴⁵, which use different molecular scores or HPO-based scores. All tested combinations showed improvements in the top-k-accuracies and are discussed in the Supplementary Material.

This comparison only focused on analyzing the performance gain by GestaltMatcher. We are not benchmarking the existing methods to find the best combination. Further parameter tuning or exploring integration methods for different scoring approaching are required in the future and can be done by anyone interested, because GestaltMatcher Code is open-source and training data is FAIR.

PEDIA main cohort

PEDIA validation cohort

Supplementary Figure 8: GestaltMatcher scores improve the accuracy of prioritization tools. Bootstrapped 95% confidence intervals are indicated by the lighter shading around the lines. GM: GestaltMatcher

Reviewer #1: Add percentages to diagnostic yield numbers in the abstract.

Response: Done.

Reviewer #2: The ClinVar ultra-rare analysis is interesting, but I think it was done by number of variants per gene rather than number of submissions per gene (though I am not entirely sure from the methods). It's not been established that disease prevalence is correlated with number of pathogenic variants, as this does not take into account the prevalence of these variants in the general population (with inheritance mode). Additionally, some of the genes with the most pathogenic variants may be from population screening of genes on the secondary findings list (who may not all develop disease) rather than from testing on patients with rare disease. This approach of counting unique variants will make the gain of function diseases that have a smaller repertoire of pathogenic variants appear more rare – and also a lab may submit one variant classification but have seen the variant multiple times. On the other hand, the decade of gene-disease relationship discovery should be more robust. It's an interesting if flawed approach that could be improved but not completely corrected. The approach is perhaps better applied as a comparison method, like they also use it here to the 2020 Turro et al. cohort (with a 16% diagnosis rate from genome sequencing). The authors attribute this result of seeing rarer conditions in this Schmidt et al. cohort to the benefit of MDT involvement rather than other differences between the cohort (prior testing, family structure, etc.). While there are many ultra-rare conditions identified in this cohort, one might expect to also see more common conditions, so the lack of more common conditions raises questions on why they are not present.

Response: We thank the reviewer for the thoughts and agree, that our visualization of count of submissions in Figure 4 is not an established proxy for prevalence and might be prone to over- and undercounting due to biases as pointed out by the reviewer. The ClinVar ultra-rare analysis already used the number of submissions of (likely) pathogenic variants per gene. We agree that the legend of Figure 4 was not stating this precisely and we therefore modified the figure legend. Additionally, we tried to improve the statistics by removing variant submissions to genes that were screened for secondary findings (59 genes of the ACMG SF v2 list were removed). The changes in Supplementary Figure 4 b and c are shown in the figure below.

For comparing variant submissions in dominant and recessive genes we do not see a simple solution in a single figure, since the information whether a variant was observed in compound heterozygous or homozygous state is not completely available from ClinVar. Likewise, the proportion compound heterozygous vs homozygous varies greatly with the autozygosity and is not reported.

In contrast to Turro, *et al.* for most participants in our study exome analysis was not the first-tier test and they had e.g. gene panel testing prior to exome sequencing. This is because in the German health-care system exome analysis was not reimbursed prior to this study in contrast to e.g. small gene panels or chromosomal microarray. This also explains why we hardly identified pathogenic variants in genes such as *BMPR2*, *EIF2AK4*, *ABCH4*, and *USH2A* as a high diagnostic yield for patients exhibiting the clinical features pulmonary arterial hypertension,

intrahepatic cholestasis, or hearing impairment can be achieved with relatively small gene panels. Variants in these genes were among the most frequently reported in Turro, *et al.*

We added these considerations to the discussion:

In our opinion, this accumulation of ultra-rare diagnoses and the relative absence of more common conditions are explained by two factors. First, many of these could be diagnosed by the broad availability of certain genetic tests, such as small gene panels or molecular karyotyping in the German healthcare system prior to this study. Second, due to the availability of these tests, the MDTs guided the diagnostic process and also could recommend targeted approaches.

Reviewer #2: YieldPred is a statistical framework for predicting the likelihood of a molecular diagnosis based on clinical features. It is not clear how this tool is contributing to the goals of this project though this dataset was used for training and testing the method. It is unclear how robust the tool is to HPO term entry. The validity of the model was not well established here.

Response: We agree with the reviewer that it is debatable whether YieldPred is required in this work. However, since Reviewer #1 saw clinical utility for expectation management, we conducted additional experiments. By training on additional cases from Turro, *et al.* we achieved a higher generalizability of the Lasso model, which is underlying YieldPred and make this model now also available at the website. For further details of the model performance we refer also to our extensive answer to the first comment of Reviewer #1.

Reviewer #2: Are a few HPO terms driving the signal?

Response: The Lasso analyses were conducted to identify the features that were positively or negatively associated with the diagnostic yield. The coefficient paths of the HPO-subgroups are shown in Figure 5 A and were checked for plausibility. E.g. cases that feature ataxia have a high chance for receiving a molecular diagnosis with ES, whereas individuals with autism have a lower chance of a conclusive test result.

Additionally, we evaluated the discriminatory power of single HPO terms. There were 1,649 unique HPO terms annotated in the TRANSLATE NAMSE cohort. Considering each HPO term separately to discriminate between solved and unsolved cases led to an average AUC of 0.5, i.e. no discriminatory power. The maximum achieved AUC of a single HPO term, namely HP:0001263 (global developmental delay), was 0.58. As a sensitivity analysis, we then fitted a logistic regression on the complete TNAMSE cohort with the top 5 HPO terms, namely HP:0001263 (global developmental delay), HP:0000252 (microcephaly), HP:0001252 (hypotonia), HP:0001250 (seizure) and HP:0001251 (ataxia), and achieved an AUC of 0.64 (95%-CI: 0.61-0.67). On the complete TNAMSE set (i.e. training and test set combined) our YieldPred model yielded an AUC of 0.72 (95%-CI: 0.69-0.74). In summary, there are HPO terms that have higher

discriminatory power than the majority of the HPO terms. However, the signal of YieldPred is additionally driven by the combination of multiple phenotypic features that are present in a patient.

Reviewer #2: Do predictions correlate with number of HPO terms?

Response: As suggested, we additionally evaluated more in detail how the model behaves. We evaluated how the number of HPO-terms affects the prediction of YieldPred, based on the individuals of TRANSLATE NAMSE and found a positive correlation. This seems to indicate that a monogenic cause is more likely if a patient exhibits a diverse set of clinical features. Our approach with HPO subgroups assures that multiple lower order terms are only counted once (e.g. brachydactyly of multiple fingers would be one feature).

Figure 5: The count of HPO-terms that was used to describe a case is correlated with the diagnostic yield. Red squares indicate the observed diagnostic yield among the cases with the respective number of HPO terms with number of cases given as text close to the squares, respectively.

Reviewer #2: The utility of the YieldPred in this study is not well described. Was YieldPred used to identify cases for reanalysis, additional functional studies (RNAseq), something else? Related, I did not understand this sentence in the discussion: “Low scores in YieldPred despite a high likelihood of a monogenic cause, e.g. a strong family history, could justify the selection of an even more comprehensive test, such as long-read genome sequencing.”

Response: We thank the reviewer for pointing out that a clearer description of the utility of YieldPred during and after the study is required. As the model was only trained on data collected during TRANSLATE NAMSE, the results of YieldPred had no direct consequence for the testing strategy in the present study. However, in future projects we could imagine a potential use of the approach. If the constellation of features as assessed by clinical geneticists is indicative of a monogenic disease but the predicted probability of a molecular diagnosis by exome sequencing is low, then subsequent testing by WGS or RNAseq might be justified. This is especially important as different diagnostic tests such as ES, short-read and long-read genome sequencing will be available necessitating algorithms to define the most reasonable testing strategy. We now elaborate on this potential application of YieldPred in the discussion:

The machine learning model YieldPred was developed to identify features that had a major impact on the diagnostic yield in our and other study cohorts. Prospectively, this approach can also be used for two purposes. Firstly, it can be used to estimate the probability that ES will result in a molecular diagnosis in each patient with a suspected rare disease and can by that means help to

manage expectations. Secondly, as YieldPred in its current form provides an estimation of the diagnostic yield of ES and not of an underlying monogenic condition of a certain individual, it can be used to stratify individuals for more comprehensive genetic testing. I.e. a low YieldPred score despite a high likelihood of a monogenic disease indicates that transcriptomics, proteomics or genome sequencing could be promising.

Reviewer #2: “Diagnoses with causal therapeutic implications” describes 5 patients (0.3% cohort, 1% diagnosed cohort) with specific targeted therapy though not a cure for all (2 were ketogenic diet). Changes in medical management are not described and may have not been systematically assessed but seems warranted to mention that diagnoses resulted in additional management changes, if that is the case.

The PEDIA tool was also described more in terms of how well it works rather than being used to diagnose additional cases in this cohort, though one case involving *KANSL* was detailed.

Response: We agree with the reviewer that the percentage of cases in which diagnoses had implications on management are low and only exemplary.

In order to highlight that cases were only exemplary and not representative, we have changed the wording accordingly:

Exemplary diagnoses with targeted therapy. Implications of diagnoses on clinical management were not assessed in a structured way. However, for five patients in the TRANSLATE-NAMSE cohort with a molecular diagnosis (1%), individualized treatments, or therapies directed against the mechanism of the disease could be initiated.

Reviewer #2: “Diagnostic yield of ES”: please clarify if this referred to only pathogenic and likely pathogenic variants or also included high interest variants of uncertain significance to be clear how the diagnostic rate is calculated.

Response: We defined four different states of a case, that are “solved”, “partially solved”, “unclear”, and “unsolved”. The label solved is only assigned if we identified pathogenic or likely pathogenic variants that could cause the described phenotypic features of the patient.

There was a single case (case ID 242) in our cohort which we assessed as partially solved. This case featured thrombocytopenia, failure to thrive, muscular hypotonia, seizures, apnea, dysphagia, and talipes. We identified a *de novo* variant in *RUNX1*: NM_001754.4:c.590_597del,

but were not sure whether the talipes could also be explained by this variant or were due to a second molecular cause that we did not identify. All solved cases, as well as the partially solved case contributed to the diagnostic yield.

In cases with the status “unclear” we identified either a VUS in a known disease gene that would match the phenotype, or variants in a candidate gene for which the evidence for a diagnostic disease gene is not sufficient yet.

All remaining cases were classified as unsolved. Unclear and unsolved cases did not contribute to the diagnostic yield.

We have now explicitly stated this also in the results section:

A molecular diagnosis was established in a total of 499 of the 1,577 patients (32%), i.e., in these cases, ES identified pathogenic or likely pathogenic variants that fully or partially explained the phenotype.

Reviewer #2: VUS are also called variants of “unknown significance” in the text which is not the typical term used in the field.

Response: Thanks for pointing this out. We replaced “unknown significance” by “uncertain significance”.

Reviewer #2: “guidelines of the American College of Medical Genetics and Genomics (ACMG) and the Association for Molecular Pathologists (AMP) for classification of sequence variants”

Response: Thanks for pointing this out. We completed and added the Association for Molecular Pathologists (AMP).

Reviewer #2: Secondary findings are described in the supplement but could warrant brief mention in the main manuscript.

Response: Thanks for the suggestion. We added the following two sentences to the section “Dual molecular diagnoses”, which we renamed to “Dual molecular diagnoses and secondary findings”:

In 17 individuals who had consented to being informed about secondary findings, we identified medically actionable variants that were unrelated to the present phenotype. The list of actionable genes was based on the ACMG recommendations, however, variants in seven additional genes were reported following discussions within the respective MDTs (Supplementary Material).

Reviewer #2: While homozygous variants were more likely to be the diagnosis in cases with consanguinity, it is worth noting that 25% of these cases had another mode of inheritance, highlighting the importance of looking more broadly.

Response: We agree, although homozygous variants were more likely to be the diagnosis in cases with consanguinity, estimating the likelihood for this mode of inheritance depends on more than just the degree of autozygosity. We modified the first sentence of the paragraph “De novo variants and parental mosaicism”:

A total of 228 diagnoses (45% of 510 diagnoses) were attributable to de novo variants, making them the most common cause of disease in families with an autozygosity below 0.02 and the second most common cause in families with consanguinity (Figure 3a, c).

We also added a sentence at the end of the paragraph about recessive disease burden referring to population genetics considerations:

However, it also has to be acknowledged that population expansion results in a drop in the prevalence of recessive disorders in random mating populations and that the lower recessive disease burden might only be a transient effect. doi:10.1002/ajmg.a.63452

Reviewer #2: The reference numbering is off in the manuscript and needs to be addressed.

Response: Thanks for pointing this out. We fixed it.

Reviewer #2: Data availability: It is not clear from the manuscript or the referenced study website if there is an ability for researchers to obtain access to the CRAMs and vcf files from the project and how this access is governed.

Response: Unfortunately, we do not have the written consent to share the original CRAM files of all cases, and thus the data availability statement was misleading. For certain subsets, such as, the PEDIA cohort, we may share vcf files and we corrected the statement:

The corresponding author agrees to fulfill any requests for materials not included in the extended data or Supplementary information, subject to verification that the request adheres to the consent provided by the research participants.

Reviewer #3: It is unclear how the 1577 patients reported here were selected from the bigger cohort of 5652 participants.

Response: A total of 5652 participants from 10 rare disease centers who did not yet have a definitive diagnosis were included in the TRANSLATE-NAMSE study. Medical history of all individuals was reviewed and discussed within multidisciplinary teams (MDTs). In the majority of cases no genetic workup was necessary e.g. because of atypical presentations of common or psychosomatic disorders. Only if a rare monogenic cause seemed probable, cases were referred to exome sequencing. We complemented the text accordingly:

The present analyses were performed using the data of a total of 1,577 of these 5,652 patients (268 adults and 1,309 children). In these individuals the MDT at the respective CRD considered a genetic cause as plausible and ES as the most suitable test (ES cohort, Supplementary Table 1).

Reviewer #3: The manuscript would benefit from more detail on how variants were prioritized for curation and reporting apart from having experienced clinicians/scientists involved after variant annotation. Typically, multiple tools, parameters and analysis strategies are used by diagnostic laboratories to ensure appropriate variants are reported and curated. For example, what proportion of patients were analysed as singletons vs trios? Were virtual panels applied based on phenotype? Which variant types exactly were included in analysis? If CNVs were systematically searched for, then what tools were used? which variant types were not systematically searched for? e.g. structural variants, UPD, mosaic?

Response: Indeed, multiple tools / parameters and analysis strategies were used and we now more precisely describe the analysis pipelines in the sections “DNA sequencing” and “ES data-processing pipeline” of the Online Methods. Additionally, we created a Supplementary Table 4 to provide the procedures performed by each laboratory. The proportion of singletons vs trios differed for the five labs and over the time span of the project, partially due to a different distribution over the disease groups and partially due to different workflows: 1) sequence trio **if** samples of parents were available, 2) sequence trio **as soon as** samples of parents became available, 3) sequence trio **only if** a diagnosis could not be established by singletons. While we did not systematically compare the diagnostic efficiency between singleton versus trio analysis our data seem to confirm results from Tan, *et al.* that showed comparable diagnostic yield but better scalability for trio analysis (Tan et al. 2019).

Reviewer #3: The additional advantage of using PEDIA needs to be benchmarked more appropriately. I am not aware of any diagnostic laboratory that solely uses CADD score or other in-silico tools to prioritize variants. There are multiple tools designed to prioritise variants based on a combination of HPO terms and variant properties (e.g. Exomiser) and a selection of these would be a more appropriate comparator.

Response: The PEDIA workflow aims at improving variant prioritization by including results of image analysis for any combination of HPO terms and variant properties. We benchmarked this additional value by investigating the top-k accuracy for four commonly used tools, that is Exomiser, Lirical, Xrare, and Amelie:

PEDIA main cohort

Reviewer #3: While I agree that the MDT approach to patient selection is well recognised to increase diagnostic yield and I am not surprised it enriched the cohort for ultra-rare diagnoses, it is also important to recognise that over time, particularly after the establishment phase, it can act as a barrier to access and to scalability. Given the large body of literature already published about the high diagnostic yield in specific patient groups (e.g. severe intellectual disability, infantile

epilepsy), a guideline-based approach may be more appropriate with MDT-based patient selection at academic centres reserved for more unusual clinical presentations and/or after an initial exome test is uninformative.

Response: We acknowledge that the MDTs might be considered as a barrier to the scalability of genetic testing. However, the practical experience showed that for most cases where exome was initiated the MDT decided quickly about the further procedure, particularly because ES is included in an increasing number of guidelines - a development which we also welcome. Therefore, the main task of the MDT is shifting to variant interpretation, discussion of additional tests that might clarify VUS, as well as identifying therapeutic options. By that means, MDTs will fulfill a similar purpose for patients with rare disorders as molecular tumor boards already do for cancer patients. We would also like to point out that another aspect of the study, which we don't report about in this manuscript, was a health economic analysis. Although already implemented in the healthcare system in other countries, we had to proof cost-effectiveness for exome-sequencing and MDTs served as a potential gate-keeper to advanced and expensive diagnostic tests. In fact, a similar framework will be used again in a new study that aims at genome diagnostics (Modellvorhaben Genomsequenzierung). We added these thoughts to a restructured discussion.

- Birgmeier, Johannes, Maximilian Haeussler, Cole A. Deisseroth, Ethan H. Steinberg, Karthik A. Jagadeesh, Alexander J. Ratner, Harendra Guturu, et al. 2020. "AMELIE Speeds Mendelian Diagnosis by Matching Patient Phenotype and Genotype to Primary Literature." *Science Translational Medicine* 12 (544). <https://doi.org/10.1126/scitranslmed.aau9113>.
- Hsieh, Tzung-Chien, Hellen Lesmann, and Peter M. Krawitz. 2023. "Facilitating the Molecular Diagnosis of Rare Genetic Disorders Through Facial Phenotypic Scores." *Current Protocols* 3 (10): e906.
- Hsieh, Tzung-Chien, Martin A. Mensah, Jean T. Pantel, Dione Aguilar, Omri Bar, Allan Bayat, Luis Becerra-Solano, et al. 2019. "PEDIA: Prioritization of Exome Data by Image Analysis." *Genetics in Medicine: Official Journal of the American College of Medical Genetics* 21 (12): 2807–14.
- Li, Qigang, Keyan Zhao, Carlos D. Bustamante, Xin Ma, and Wing H. Wong. 2019. "Xrare: A Machine Learning Method Jointly Modeling Phenotypes and Genetic Evidence for Rare Disease Diagnosis." *Genetics in Medicine: Official Journal of the American College of Medical Genetics* 21 (9): 2126–34.
- Retterer, Kyle, Jane Jussola, Megan T. Cho, Patrik Vitazka, Francisca Millan, Federica Gibellini, Annette Vertino-Bell, et al. 2016. "Clinical Application of Whole-Exome Sequencing across Clinical Indications." *Genetics in Medicine: Official Journal of the American College of Medical Genetics* 18 (7): 696–704.
- Robinson, Peter N., Sebastian Köhler, Anika Oellrich, Sanger Mouse Genetics Project, Kai Wang, Christopher J. Mungall, Suzanna E. Lewis, et al. 2014. "Improved Exome Prioritization of Disease Genes through Cross-Species Phenotype Comparison." *Genome Research* 24 (2): 340–48.

- Robinson, Peter N., Vida Ravanmehr, Julius O. B. Jacobsen, Daniel Danis, Xingmin Aaron Zhang, Leigh C. Carmody, Michael A. Gargano, et al. 2020. "Interpretable Clinical Genomics with a Likelihood Ratio Paradigm." *American Journal of Human Genetics* 107 (3): 403–17.
- Sobrido, María-Jesús, Peter Bauer, Tom de Koning, Thomas Klopstock, Yann Nadjar, Marc C. Patterson, Matthis Synofzik, and Chris J. Hendriksz. 2019. "Recommendations for Patient Screening in Ultra-Rare Inherited Metabolic Diseases: What Have We Learned from Niemann-Pick Disease Type C?" *Orphanet Journal of Rare Diseases* 14 (1): 20.
- Tan, Tiong Yang, Sebastian Lunke, Belinda Chong, Dean Phelan, Miriam Fanjul-Fernandez, Justine E. Marum, Vanessa Siva Kumar, et al. 2019. "A Head-to-Head Evaluation of the Diagnostic Efficacy and Costs of Trio versus Singleton Exome Sequencing Analysis." *European Journal of Human Genetics: EJHG* 27 (12): 1791–99.

Decision Letter, first revision:

9th February 2024

Dear Peter,

Your revised manuscript "Next-generation phenotyping integrated in a national framework for patients with ultra-rare disorders improves genetic diagnostics and yields new molecular findings" (NG-A63044R) has been seen by the original referees. As you will see from their comments below, they find that the paper has improved in revision, and therefore we will be happy in principle to publish it in Nature Genetics as an Article pending final revisions to address Reviewer #2's remaining requests and to comply with our editorial and formatting guidelines.

We are now performing detailed checks on your paper, and we will send you a checklist detailing our editorial and formatting requirements soon. Please do not upload the final materials or make any revisions until you receive this additional information from us.

Thank you again for your interest in Nature Genetics. Please do not hesitate to contact me if you have any questions.

Sincerely,
Kyle

Kyle Vogan, PhD
Senior Editor
Nature Genetics
<https://orcid.org/0000-0001-9565-9665>

Reviewer #1 (Remarks to the Author):

I appreciate the additional analysis that the authors have performed based on the feedback that I and

the other reviewers provided. The paper is much improved now.

On the additional analysis of YieldPred on other datasets: if the goal was to assess YieldPred on a more balanced, less neurological heavy, cohort I am not sure the NIHR BioResource cohort was an ideal choice as it has its own heavy bias towards haematological disorders that are the focus of their Addenbrookes Hospital based research group. Therefore, I don't think it really highlights differences between the German and UK healthcare systems as claimed. To achieve this, it would have been better to compare to the 100,000 Genomes Project or Genomic Medicine Service data made available in the Genomics England research environment, with a much larger cohort of HPO annotated patients that were recruited across most types of rare disease and across the whole of NHS England. I suspect the drop in performance observed in the Cambridge cohort would not have been as dramatic. However, I am happy that they have investigated the potential bias of the original YieldPred tool and produced an improved model. A separate, future study that looks at the application to other large-scale disease sequencing projects would definitely be welcomed by the community.

The authors have added a very impressive analysis of PEDIA with the 4 tools I suggested on the cohort(s) and the performance gain when GestaltMatcher is included is clear, showing the potential of these image-based approaches.

Reviewer #2 (Remarks to the Author):

The authors addressed my prior concerns and have made nice additions to the manuscript. There are some limitations but overall it is a well conducted study that will be of high interest to the field.

For the estimate that roughly 80% of 5000 rare diseases have a prevalence of <1 in 1 million, is this restricting to genetic rare diseases (as I've heard that estimate that ~80% of rare diseases are genetic)? May want to clarify this (one option is just to include the word genetic on line 160).

Line 384 – LIRICAL should be capitalized.

The data access section is still misleading and should be clearer that the sequence data was not consented for sharing except for a small subset of it that may be available on request.

Reviewer #3 (Remarks to the Author):

Thank you for the opportunity to review this manuscript again. The authors have addressed my previous comments/suggestions satisfactorily.

Author Rebuttal, first revision:

Reviewer #1: I appreciate the additional analysis that the authors have performed based on the feedback that I and the other reviewers provided. The paper is much improved now.

On the additional analysis of YieldPred on other datasets: if the goal was to assess YieldPred on a more balanced, less neurological heavy, cohort I am not sure the NIHR BioResource cohort was an ideal choice as it has its own heavy bias towards haematological disorders that are the focus of their Addenbrookes Hospital based research group. Therefore, I don't think it really highlights differences between the German and UK healthcare systems as claimed. To achieve this, it would have been better to compare to the 100,000 Genomes Project or Genomic Medicine Service data made available in the Genomics England research environment, with a much larger cohort of HPO annotated patients that were recruited across most types of rare disease and across the whole of NHS England. I suspect the drop in performance observed in the Cambridge cohort would not have been as dramatic. However, I am happy that they have investigated the potential bias of the original YieldPred tool and produced an improved model. A separate, future study that looks at the application to other large-scale disease sequencing projects would definitely be welcomed by the community.

The authors have added a very impressive analysis of PEDIA with the 4 tools I suggested on the cohort(s) and the performance gain when GestaltMatcher is included is clear, showing the potential of these image-based approaches.

Response: We thank the reviewer for their thoughts and valuable feedback. We agree that the potential of YieldPred would be of interest to the community and requires further investigation in a separate study.

Reviewer #2: The authors addressed my prior concerns and have made nice additions to the manuscript. There are some limitations but overall it is a well conducted study that will be of high interest to the field.

For the estimate that roughly 80% of 5000 rare diseases have a prevalence of <1 in 1 million, is this restricting to genetic rare diseases (as I've heard that estimate that ~80% of rare diseases are genetic)? May want to clarify this (one option is just to include the word genetic on line 160).

Response: We thank the reviewer for their positive feedback. We adapted the sentence accordingly, it now reads

*It is estimated that roughly 80% of the more than 5000 rare **genetic** diseases have a prevalence below one in a million.*

Reviewer #2: Line 384 – LIRICAL should be capitalized.

Response: We thank the reviewer for pointing that out and corrected the spelling throughout the manuscript and the Supplementary Material, including the figures.

Reviewer #2: The data access section is still misleading and should be clearer that the sequence data was not consented for sharing except for a small subset of it that may be available on request.

Response: We apologize for any confusion about the data availability statement and adapted it accordingly:

The corresponding author agrees to fulfill any requests for materials not included in the extended data or Supplementary Material, subject to verification that the request adheres to the consent provided by the research participants. Patient-related data not included in the article may be subject to patient confidentiality. Raw sequencing data was not consented for sharing, except for the PEDIA subset which is available upon request.

Reviewer #3: Thank you for the opportunity to review this manuscript again. The authors have addressed my previous comments/suggestions satisfactorily.

Response: We thank the reviewer for their positive feedback.

Final Decision Letter:

18th June 2024

Dear Peter,

I am delighted to say that your manuscript "Next-generation phenotyping integrated in a national framework for patients with ultra-rare disorders improves genetic diagnostics and yields new molecular findings" has been accepted for publication in an upcoming issue of Nature Genetics.

Your paper will be published online after we receive your corrections and will appear in print in the next available issue. You can find out your date of online publication by contacting the Nature Press Office (press@nature.com) after sending your e-proof corrections.

Before your paper is published online, we will be distributing a press release to news organizations worldwide, which may very well include details of your work. We are happy for your institution or funding agency to prepare its own press release, but it must mention the embargo date and Nature Genetics. Our Press Office may contact you closer to the time of publication, but if you or your Press Office have any enquiries in the meantime, please contact press@nature.com.

Please note that Nature Genetics is a Transformative Journal (TJ). Authors may publish their research with us through the traditional subscription access route or make their paper immediately open access through payment of an article-processing charge (APC). Authors will not be required to make a final decision about access to their article until it has been accepted. Find out more about Transformative Journals

Authors may need to take specific actions to achieve compliance with funder and institutional open access mandates. If your research is supported by a funder that requires immediate open access (e.g. according to Plan S principles), then you should select the gold OA route, and we will direct you to the compliant route where possible. For authors selecting the subscription publication route, the journal's standard licensing terms will need to be accepted, including [a href="https://www.nature.com/nature-portfolio/editorial-policies/self-archiving-and-license-to-publish"](https://www.nature.com/nature-portfolio/editorial-policies/self-archiving-and-license-to-publish). Those licensing terms will supersede any other terms that the author or any third party may

assert apply to any version of the manuscript.

If you have not already done so, we strongly recommend that you upload the step-by-step protocols used in this manuscript to protocols.io. protocols.io is an open online resource that allows researchers to share their detailed experimental know-how. All uploaded protocols are made freely available and are assigned DOIs for ease of citation. Protocols can be linked to any publications in which they are used and will be linked to from your article. You can also establish a dedicated workspace to collect all your lab Protocols. By uploading your Protocols to protocols.io, you are enabling researchers to more readily reproduce or adapt the methodology you use, as well as increasing the visibility of your protocols and papers. Upload your Protocols at <https://protocols.io>. Further information can be found at <https://www.protocols.io/help/publish-articles>.

Sincerely,
Kyle

Kyle Vogan, PhD
Senior Editor
Nature Genetics
<https://orcid.org/0000-0001-9565-9665>